# Engineering bioactive nanoparticles to rejuvenate vascular progenitor cells

Loan Bui[1], Shanique Edwards[2], Eva Hall[1], Laura Alderfer[1], Kellen Round [1], Madeline Owen[1], Pietro Sainaghi[1], Siyuan Zhang [3,4], Prakash D. Nallathamby[1], Laura S. Haneline[2,5,6] & Donny Hanjaya-Putra [1,4,7,8✉]

Fetal exposure to gestational diabetes mellitus (GDM) predisposes children to future health complications including type-2 diabetes mellitus, hypertension, and cardiovascular disease. A key mechanism by which these complications occur is through stress-induced dysfunction of endothelial progenitor cells (EPCs), including endothelial colony-forming cells (ECFCs). Although several approaches have been previously explored to restore endothelial function, their widespread adoption remains tampered by systemic side effects of adjuvant drugs and unintended immune response of gene therapies. Here, we report a strategy to rejuvenate circulating vascular progenitor cells by conjugation of drug-loaded liposomal nanoparticles directly to the surface of GDM-exposed ECFCs (GDM-ECFCs). Bioactive nanoparticles can be robustly conjugated to the surface of ECFCs without altering cell viability and key progenitor phenotypes. Moreover, controlled delivery of therapeutic drugs to GDM-ECFCs is able to normalize transgelin (TAGLN) expression and improve cell migration, which is a critical key step in establishing functional vascular networks. More importantly, sustained pseudo-autocrine stimulation with bioactive nanoparticles is able to improve in vitro and in vivo vasculogenesis of GDM-ECFCs. Collectively, these findings highlight a simple, yet promising strategy to rejuvenate GDM-ECFCs and improve their therapeutic potential. Promising results from this study warrant future investigations on the prospect of the proposed strategy to improve dysfunctional vascular progenitor cells in the context of other chronic diseases, which has broad implications for addressing various cardiovascular complications, as well as advancing tissue repair and regenerative medicine.

[1] Department of Aerospace and Mechanical Engineering, Bioengineering Graduate Program, University of Notre Dame, Notre Dame, IN 46556, USA. [2] Herman B Wells Center for Pediatric Research, Department of Pediatrics, Indiana University School of Medicine, Riley Hospital for Children at Indiana University Health, Indianapolis, IN 46202, USA. [3] Department of Biological Science, University of Notre Dame, Notre Dame, IN 46556, USA. [4] Harper Cancer Research Institute, University of Notre Dame, Notre Dame, IN 46556, USA. [5] Department of Microbiology and Immunology, Indiana University School of Medicine, Indianapolis, IN 46202, USA. [6] Department of Anatomy, Cell Biology, and Physiology, Indiana University School of Medicine, Indianapolis, IN 46202, USA. [7] Department of Chemical and Biomolecular Engineering, University of Notre Dame, Notre Dame, IN 46556, USA. [8] Center for Stem Cells and Regenerative Medicine, University of Notre Dame, Notre Dame, IN 46556, USA. ✉email: dputra1@nd.edu

Cardiovascular disease is the most prevalent cause of mortality and morbidity among patients with diabetes[1]. Adults with diabetes have a two to six times higher risk of developing cardiovascular disease than unaffected individuals[2]. Similarly, fetal exposure to gestational diabetes mellitus (GDM), which affects 6–15% of all pregnancy, predisposes children to future health complications including type 2 diabetes mellitus (T2DM), hypertension, and cardiovascular disease[3–5]. Although the pathophysiology that links diabetes and cardiovascular disease is complex and multifactorial, there is a general agreement that hyperglycemia and oxidative stress lead to stress-induced early endothelial dysfunction[6], which is responsible for both macrovascular (i.e., peripheral artery disease, stroke) and microvascular (i.e., diabetic nephropathy, retinopathy) complications[7–9]. Several pre-clinical and clinical trials are exploring the therapeutic effect of stem and progenitor cell therapies to repair the damaged endothelium and promote neovascularization[10–12]. One promising autologous cell source is endothelial colony-forming cells (ECFCs), a subtype of endothelial progenitor cells (EPCs), identified from circulating adult blood and highly enriched in human umbilical cord blood[13,14]. As putative EPCs, these ECFCs express robust proliferative potential in forming secondary and tertiary colonies, as well as de novo blood vessel formation in vivo[13]. Nonetheless, hyperglycemia and a diabetic intrauterine environment also cause premature senescence and significant dysfunction of ECFCs, which limit their therapeutic use[3,15,16]. ECFCs isolated from patients with diabetes demonstrate delayed colony formation, reduced cell migration, and impaired vasculogenic potential[3,15,17]. Therefore, restoring dysfunctional ECFCs could improve their vasculogenic potential and serve as biomarkers to assess cardiovascular disease risk[10,11].

Several approaches have been explored to rejuvenate ECFCs and to restore their therapeutic potential. These approaches include the delivery of adjuvant drugs to improve immobilization of progenitor cells, genetic modification of cells to overexpress growth factors, and pre-conditioning of these cells with pharmacological agents[18–21]. Nonetheless, adjuvant agents need to be maintained at high and sustained systemic levels for efficacy, while genetically modified cells pose high regulatory and cost barriers, which altogether hinder their clinical implementation[22]. Efforts to enhance the stability and effective presentation of bioactive molecules for improving the therapeutic potential of ECFCs have included attempts to judiciously conjugate vascular endothelial growth factor (VEGF) onto the surface of microparticles[23], as well as presentation of proteins and glycomimetic agents to the cell surface[17,18,24,25]. Despite recent progress in these areas[26], translational challenges persist for rejuvenating EPCs using growth factor and gene therapies, including unintended immune response, enzymatic degradation, and uncertain toxicology[27–29].

We report a class of liposomal nanoparticles with tunable release kinetics, which can be conjugated directly onto the cell surface to improve the therapeutic potential of ECFCs isolated from infants born to women with gestational diabetes mellitus (GDM-ECFCs). This work builds on the discovery that GDM-ECFCs have decreased vasculogenic potential and altered gene expression, particularly transgelin (TAGLN)[3,30], also known as smooth muscle protein 22α (SM22α). We previously reported that increased TAGLN expression in GDM-ECFCs is associated with disrupted actin cytoskeletal rearrangement, which results in reduced cell migration and impaired vasculogenesis[3,30,31]. Since TAGLN is a direct target of the TGF-β/Smad3 pathway[32], we hypothesized that delivering TGF-β inhibitor (SB-431542) directly to the surface of the cells could normalize TAGLN expression and eventually improve cell migration, which is a critical key step in establishing functional vascular networks.

Liposomal formulation of nanoparticles was selected for the fabrication technique because it can form multilamellar structures with high encapsulation efficiency and exhibits controllable release kinetics[33,34]. Moreover, since a similar formulation of liposomal nanoparticles has been previously used to deliver adjuvant drugs to hematopoietic stem and progenitor cells (HSPCs) without adverse immune response[22,35], the utilization of liposomal nanoparticles can ease the clinical translation of our strategy [36].

We demonstrate that such lipid-based nanoparticles can robustly bind onto the surface of GDM-ECFCs without altering cell viability and key ECFC phenotypes. More importantly, bioactive nanoparticles can significantly normalize TAGLN expression, restore cell migration, as well as improve vasculogenesis in vitro and in vivo. Collectively, with localized and sustained pseudo-autocrine stimulation of therapeutic cells with bioactive nanoparticles, we present a potentially promising strategy to rejuvenate GDM-ECFCs and improve their therapeutic potential to regenerate the vasculature and address a range of cardiovascular complications that precipitate from GDM.

## Results

**Engineering bioactive nanoparticles to the cell surfaces.** Based on our previous findings that TAGLN inhibition can enhance the functionality of GDM-ECFCs[15,30], we designed a nanocarrier system to deliver small molecule SB-431542 (TGF-β inhibitor) for sustained pseudo-autocrine stimulation to therapeutic cells (Fig. 1a). Utilizing a lipid formulation that includes thiol-reactive maleimide headgroups[35], multilamellar lipid nanoparticles with a desired size of 150 nm in diameter were engineered to control the release of bioactive agents (Fig. 1b and Supplementary Fig. 1). The targeted size and multilamellar structures of the nanoparticles were chosen because previous studies with these similar parameters showed high encapsulation efficiencies and minimal inflammatory responses following in vivo implantation[35]. Using Dynamic Light Scattering (DLS) and Nanosight measurements, we confirmed that the multilamellar lipid nanoparticles with 147 ± 63 nm diameter were stable for at least 30 days at both 4 °C and 37 °C (Fig. 1c, d and Supplementary Fig. 2), which is suitable for long-term storage and clinical applications. To stably couple bioactive nanoparticles to the surface of ECFCs, we exploited the high level of reduced thiol groups present on the surface of many progenitor cells[37,38]. First, we detected substantial amounts of free thiols on ECFCs (Supplementary Table 1). Despite some variations between biological replicates from different patient samples, there were no significant differences between the level of free thiols among normal ECFCs and GDM-ECFCs ($p > 0.688$; Supplementary Fig. 3). Next, nanoparticles were conjugated to the surfaces of ECFCs using a benign maleimide-thiol coupling, followed by in situ PEGylation to quench residual reactive groups of the nanoparticles (Fig. 1a)[38]. Upon conjugation, the nanoparticles were successfully attached to the cell surfaces (Fig. 1e). Increasing the ratio of cells to nanoparticles from 1:100 to 1:5,000 correlates well with the increase in the mean fluorescence intensity (MFI) as detected by flow cytometry analysis (Fig. 1f). Since the MFI from samples with cells to nanoparticles ratio of 1:5,000 starts to deviate from linearity, we determined that 5,000 (±100) was the maximum number of particles that can be conjugated on the surface of ECFCs without affecting cell viability and proliferation (Fig. 1f, g and Supplementary Fig. 4). These findings are consistent with previous studies reporting that attachment of nanoparticles, each 150 nm in diameter, would occlude only 5% of the surface of a typical 20-μm-diameter cell[35,38]. Moreover, nanoparticles could be loaded with various concentrations of the

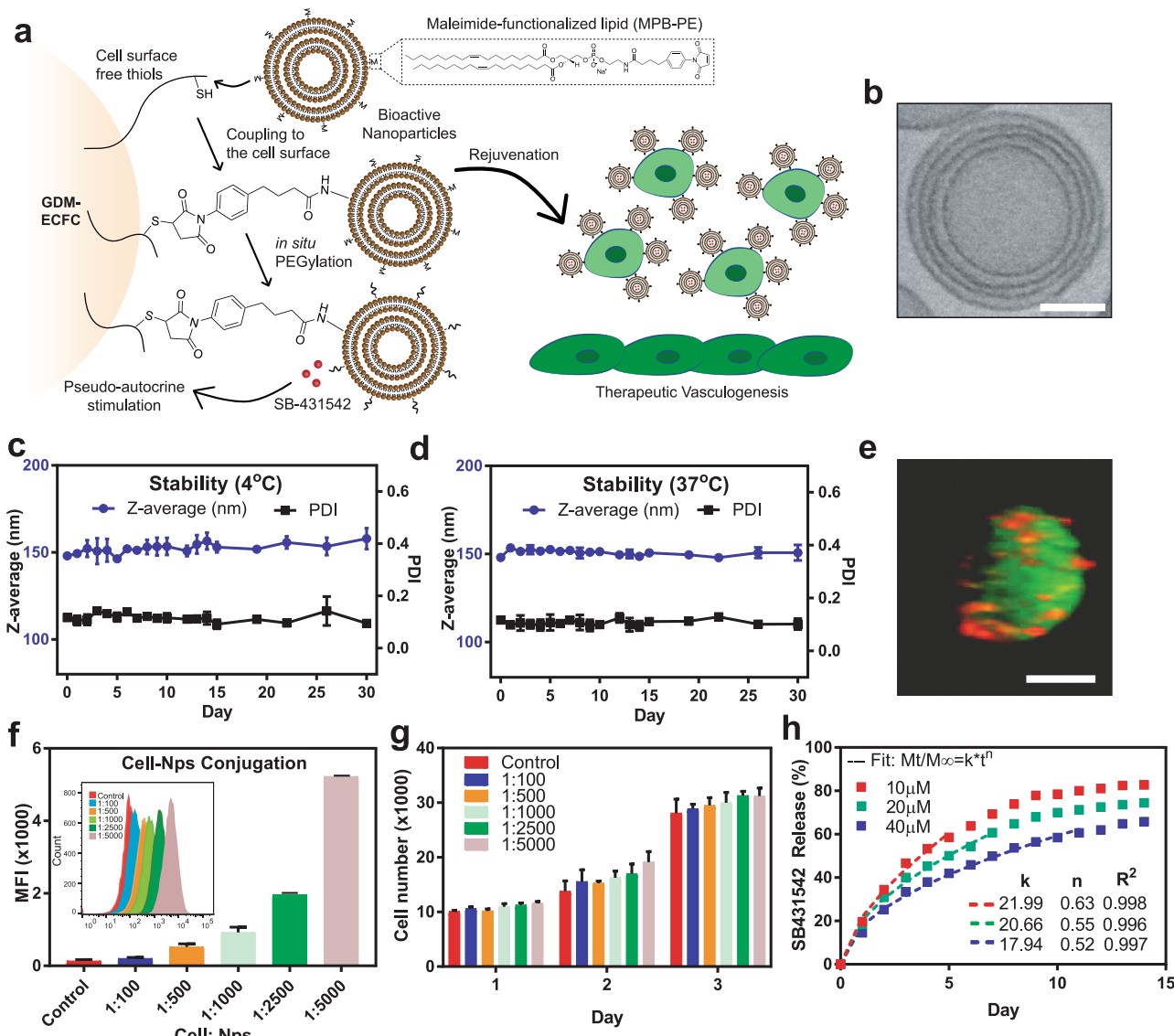

**Fig. 1 Synthesis and characterization of bioactive nanoparticles. a** Schematic of maleimide-based conjugation of bioactive nanoparticles to the ECFC surface's free thiols followed by in situ PEGylation. Continuous pseudo-autocrine stimulation of GDM-ECFCs with SB-431542 improves their clinical potential for therapeutic vasculogenesis. **b** A CryoTEM photograph demonstrating the multilamellar structure of the nanoparticle. Scale bar is 50 nm. **c, d** The stability of the nanoparticles was quantified using DLS for hydrodynamic diameter (z-average) and polydispersity (PDI) over 30 days in buffer at (**c**) 4 °C and (**d**) 37 °C (physiological temperature). The data represent the mean ± s.d. of three independent experiments conducted in triplicate. **e** A confocal photograph demonstrating a stable conjugation of Dil-labeled multilamellar lipid nanoparticles (red) conjugated onto the surface of a CFSE-labeled ECFC (green). Scale bar is 10 μm. **f** Flow cytometry analysis demonstrated an increase in mean fluorescence intensity (MFI) when the ratio of cell to nanoparticles was increased from 1:100 to 1:5,000. Inset indicates the corresponding histogram data. **g** The viability of ECFCs with various cell to nanoparticle ratios (1:100 to 1:5,000) was quantified using alamar blue assay over three days. No significant difference was observed between non-conjugated cells and nanoparticle-conjugated cells (mean ± S.D., three independent experiments conducted in triplicate). **h** Accumulative release of the bioactive small molecule, SB-431542 (SB), from the NPs over 2 weeks. Three different initial concentrations of SB (10, 20, 40 μM) were encapsulated into the NPs and the amounts of SB released were measured daily. The release profiles were fit for Korsmeyer-Peppas equation and the fitting parameters are shown within the graph.

hydrophobic small molecule (SB-431542) and demonstrated consistent week-long drug release profiles (Fig. 1h). By fitting the drug release kinetic to the Korsmeyer-Peppas model[39], $n$ values of 0.52 to 0.63 were obtained, suggesting that non-Fickian diffusion is the main driving force of drug release (Fig. 1h and Supplementary Fig. 4). Based on these findings and previous studies that reported a lack of an inflammatory response from innate immune cells exposed to the nanoparticles[35,38], we utilized these lipid multilamellar nanoparticles (up to 1:5,000 ratio) loaded with 40 μM SB-431542 for our subsequent in vitro functionality and in vivo therapeutic studies.

**Conjugation of nanoparticles does not alter ECFC phenotypes.** Once the optimum nanoparticle properties and parameters were determined, we next tested if nanoparticle conjugation altered key ECFC functions and phenotypes. ECFCs conjugated with nanoparticles up to 5000 (±100) per cell retained a robust proliferative potential that was comparable to unconjugated control cells (Fig. 2a–d). During cell division, nanoparticles attached to the surface of ECFCs segregated equally to their daughter cells, as reflected by a stepwise decrease in the MFI from ECFC-conjugated nanoparticles, corresponding with sequential divisions over several days (Fig. 2c, d and Supplementary Fig. 5).

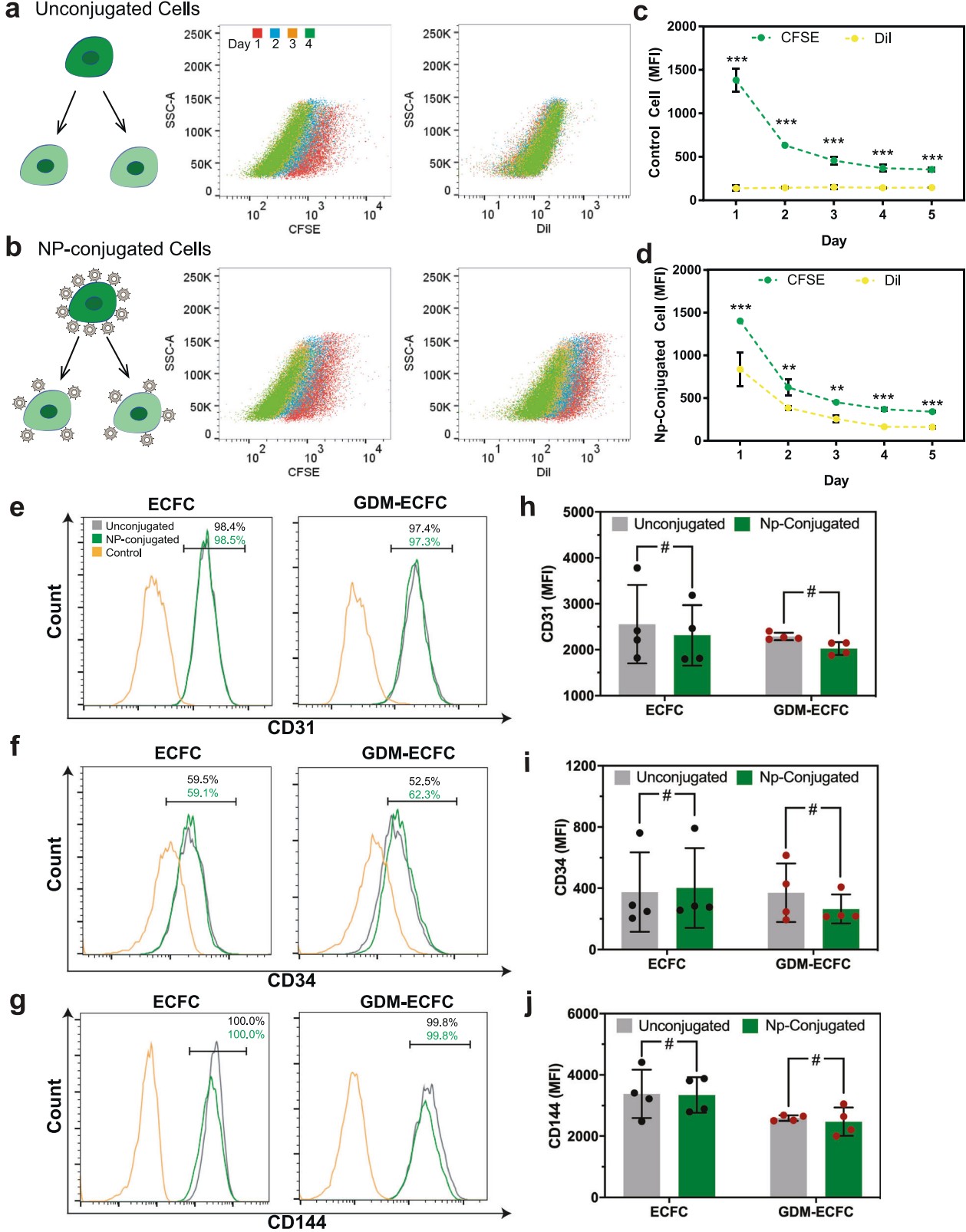

Furthermore, we also assessed the impact of nanoparticles conjugation on cell surface markers specific to ECFCs[10,13]. Comparative flow cytometry analysis revealed that expression levels of CD31, CD34, and CD144 were comparable between normal ECFCs and GDM-ECFCs (Fig. 2e–g and Supplementary Fig. 6). Despite a slight variation in CD31 expression among the biological replicates, there was no statistical difference between them ($^{\#}P = 0.387$). In addition, nanoparticle conjugation did not alter expression of CD31, CD34, and CD144 for normal ECFCs and GDM-ECFCs (Fig. 2h–j and Supplementary Fig. 6). Collectively, these data suggest that conjugation of nanoparticles does not significantly alter key functions and phenotypes of ECFCs.

**Fig. 2 Characterization of key ECFC phenotypes.** CFSE-labelled ECFCs were (**a**) unmanipulated as control unconjugated cells or (**b**) conjugated with 5000 Dil-labeled multilamellar lipid nanoparticles per cell. Scatter plots demonstrate that nanoparticles remained on the cell surface and split equally among daughter cells. Quantification of mean fluorescence intensity (MFI) of CFSE and Dil over 5 days for (**c**) control unconjugated cells and (**d**) nanoparticle-conjugated cells (mean ± s.d., three independent experiments conducted in triplicate). Expression of cell surface markers (**e**) CD31, (**f**) CD34, and (**g**) CD144 were examined using flow cytometry for NP-conjugated cells (*green* lines) and compared to unconjugated cells (*grey* lines) and isotype controls (*yellow* lines). Nanoparticle-conjugated cells and their unconjugated counterparts expressed comparable levels of ECFC-specific markers, based on their MFIs quantified for (**h**) CD31, (**i**) CD34, and (**j**) CD144. Four biological replicates (n = 4; mean ± s.d.) were used for normal ECFCs (*black* data dots) and GDM-ECFCs (*red* data dots). Statistical significance was set at #P > 0.05, **P < 0.01, ***P < 0.005. Representative histograms from individual ECFC line can be found in Supplementary Fig. 6.

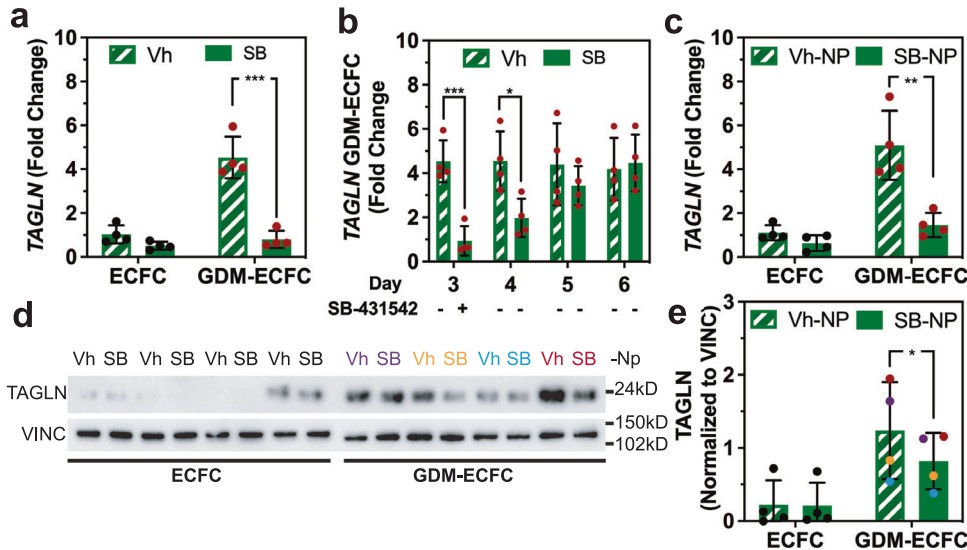

**Fig. 3 Treatment with SB-431542 reduces TAGLN expression in GDM-ECFCs. a** Real-time RT-PCR quantification of *TAGLN* expression in normal ECFC and GDM-ECFC under Vehicle (DMSO) control or treatment with 5 μM SB-431542 (SB) for 72 h (four biological replicates, n = 4; mean ± s.d.). *TAGLN* expression is normalized to ECFC. **b** The inhibition effects of SB was transient, as the SB was removed from the medium at day 4, the *TAGLN* expression in GDM-ECFC at day 5 and 6 increased to level comparable to GDM-ECFC treated with vehicle control. **c** Real-time RT-PCR quantification of *TAGLN* expression in normal ECFC and GDM-ECFC conjugated with either Vh-NPs or SB-NPs after 6 days in culture (four biological replicates, n = 4; mean ± s.d.). Bioactive SB-NPs provided a continuous down regulation of *TAGLN* at the mRNA level following 6 days in culture. **d** Representative Western blot evaluating TAGLN is shown using whole cell lysates isolated at day 6 from normal ECFC (n = 4) and GDM-ECFCs (n = 4) conjugated with either Vh-NPs or SB-NPs. Vinculin (VINC) is the loading control. **e** Band intensities were quantified using Image J and TAGLN protein expression levels were normalized using VINC. Individual data sets are color-coded (purple, yellow, blue, and red) to match the corresponding western blot bands and the GDM severity in the patient population (Supplementary Table 1). Statistical significance was evaluated using Student's t-test. Significance levels were set at *P < 0.05, **P < 0.01, ***P < 0.005.

**Drug loaded nanoparticles normalize TAGLN expression.** Because *TAGLN* is a TGF-β inducible gene that contributes to GDM-ECFC dysfunction[19], we investigated whether nanoparticles loaded with SB-431542 and conjugated to the surface of ECFCs can stably normalize *TAGLN* expression. First, we confirmed using qRT-PCR that GDM-ECFCs cultured in media supplemented with 5 μM of SB-431542 for 72 h demonstrated a decrease in the relative mRNA expression of *TAGLN* (Fig. 3a). However, the SB-431542-mediated reduction of *TAGLN* expression in GDM-ECFCs was transient and unstable (Fig. 3b). Upon removal of the SB-431542 from the culture media of GDM-ECFCs at day 4, relative *TAGLN* mRNA expression increased to levels comparable to *TAGLN* expression in vehicle control GDM-ECFCs at day 5 and 6 (Fig. 3b). These findings are consistent with previous studies that showed continuous SB-431542 supplementation in the media is required for the expansion and maintenance of ECs derived from human pluripotent stem cells (hPSCs)[40–42]. Motivated by these observations, we tested if SB-431542-loaded nanoparticles (SB-NPs) conjugated to the surface of ECFCs can provide a pseudo-autocrine stimulation to stably normalize *TAGLN* expression. Indeed, *TAGLN* mRNA levels were reduced in GDM-ECFCs conjugated with the SB-NPs compared to vehicle controls (Vh-NPs). The decrease in *TAGLN*

expression appeared to be stable for at least 6 days in culture (Fig. 3c). At the protein level, a steep decline in TAGLN expression was observed in two GDM-ECFC lines with the most severe TAGLN expression, while a more modest decline is observed in the other two GDM-ECFC lines with the least severe TAGLN expression (Fig. 3d, e). Although there were variations in TAGLN expression among biological replicates, which correlated well with GDM severity in the patient population (Supplementary Table 1), western blot analysis revealed that protein expression of TAGLN was significantly lower for GDM-ECFCs conjugated with SB-NPs compared to Vh-NPs controls (Fig. 3e and Supplementary Table 2). These results underscore the stability of *TAGLN* inhibition using SB-NPs as a critical enabling step in improving the therapeutic potential of GDM-ECFCs.

**Bioactive nanoparticles improve cell migration in vitro.** Since TAGLN is an actin binding protein implicated in regulating cell migration, a critical step in establishing vascular networks[15,30], we next investigated if SB-NPs could improve cell migration and vasculogenic potential of GDM-ECFCs. Initial investigations focused on the ability of SB-NPs to improve cell migration in trans-well migration and wound healing assays. Normalizing

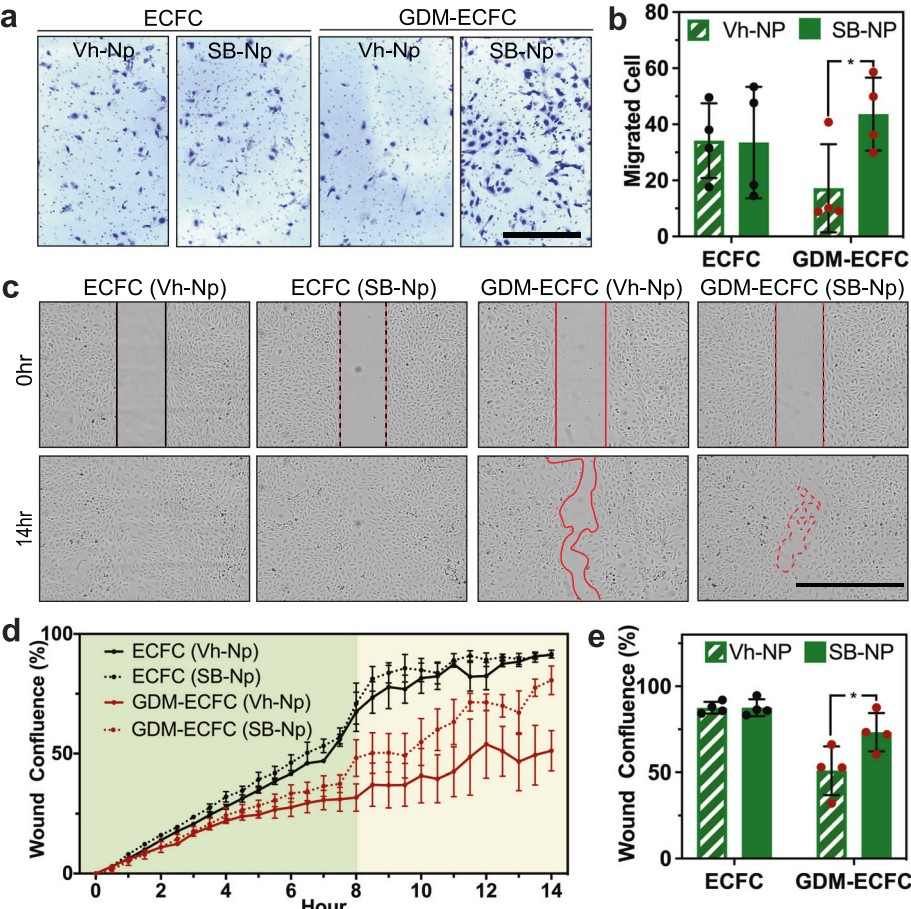

**Fig. 4 Bioactive nanoparticles improve cell migration. a** Transwell migration assays were performed with normal ECFCs and GDM-ECFCs conjugated with Vh-NPs or SB-NPs. Photomicrographs depict migrated ECFCs stained with crystal violet. Scale bar is 1 mm. **b** The number of migrating cells after 4 h were quantified. Four biological replicates ($n = 4$; mean ± s.d.) were used for normal ECFCs (*black* data dots) and GDM-ECFCs (*red* data dots). SB-NPs significantly improve cell migration of GDM-ECFCs (*$P = 0.039$), but not normal ECFCs ($P = 0.955$). **c** Wound healing assays were performed with normal ECFCs and GDM-ECFCs treated with Vh-NPs or SB-NPs. High contrast brightfield images depict migrated ECFCs at 0 h and 14 h post wound initiation. Scale bar is 1 mm. **d** Kinetic wound confluence curves indicate wound closure for normal ECFCs and GDM-ECFCs treated with Vh-NPs or SB-NPs (mean ± s.d.). **e** Quantification of wound confluence at 8 h post wound initiation. Four biological replicates ($n = 4$; mean ± s.d.) were used for normal ECFCs (*black* data dots) and GDM-ECFCs (*red* data dots). SB-NPs significantly improve wound closure of GDM-ECFCs (*$P = 0.048$), but not normal ECFCs ($P = 0.988$). Statistical significance was evaluated using Student's *t*-test. Significance levels were set at *$P < 0.05$, **$P < 0.01$, ***$P < 0.005$.

TAGLN expression in GDM-ECFCs using SB-NPs resulted in a greater number of cells migrating towards the pro-migratory stimulus (Fig. 4a and Supplementary Fig. 7). Using four biological replicates for each condition, we observed that SB-NPs significantly improve cell migration of GDM-ECFCs (*$P = 0.039$), but not normal ECFCs (#$P = 0.955$, Fig. 4b). To obtain a kinetic readout of ECFC migration, a wound healing assay was performed using a time-lapse image analysis (Fig. 4c and Supplementary Movie 1–4). Starting at 8 h after wound initiation, GDM-ECFCs conjugated with the SB-NPs demonstrated a significant improvement in wound closure, improving until at least 14 h after wound initiation, compared to GDM-ECFCs conjugated with the Vh-NPs control (Fig. 4d). Quantification of the wound closure at 8 h demonstrated that SB-NPs significantly increased wound closure by GDM-ECFCs (*$P = 0.048$), but not by normal ECFCs (#$P = 0.988$, Fig. 4e). Overall, these results suggest that SB-NPs significantly improve cell migration, which was impaired in GDM-ECFCs.

**Bioactive nanoparticles augment 2D and 3D vasculogenesis in vitro.** Our next investigations focused on evaluating if the improvement in cell migration could restore the vasculogenic potential of GDM-ECFCs in vitro. A vascular tube formation assay in 2D on Matrigel was performed for normal and GDM-ECFCs with either Vh-NPs or SB-NPs (Fig. 5a and Supplementary Movie 5–8). We utilized the kinetic analysis of vasculogenesis (KAV), which was extensively characterized in our previous studies[30,31,43,44], to quantitate large time-lapsed image data sets and provide high-throughput vasculogenic analysis. While other key parameters (e.g., tube length, vessel area) were quantified, we primarily focused on closed networks because our previous studies have identified this parameter as the most significant phenotypes effected by intrauterine GDM exposure[3,31]. Similar to previously observed biphasic trends in ECFC network formation kinetics[31,43], we confirmed an increasing number of closed networks occurred in *phase 1* (0–5 h) until network formation peaks at 5 h, and then a slight decrease in closed networks occurred in *phase 2* (5–10 h; Fig. 5b). Quantification of the closed networks at 5 h demonstrated that SB-NPs significantly increased vascular closed networks formed by GDM-ECFCs (**$P = 0.010$), but not by normal ECFCs (#$P = 0.218$) in comparison to the Vh-NPs corresponding control groups (Fig. 5c). Significantly, the numbers of closed networks in GDM-ECFCs conjugated with SB-NPs

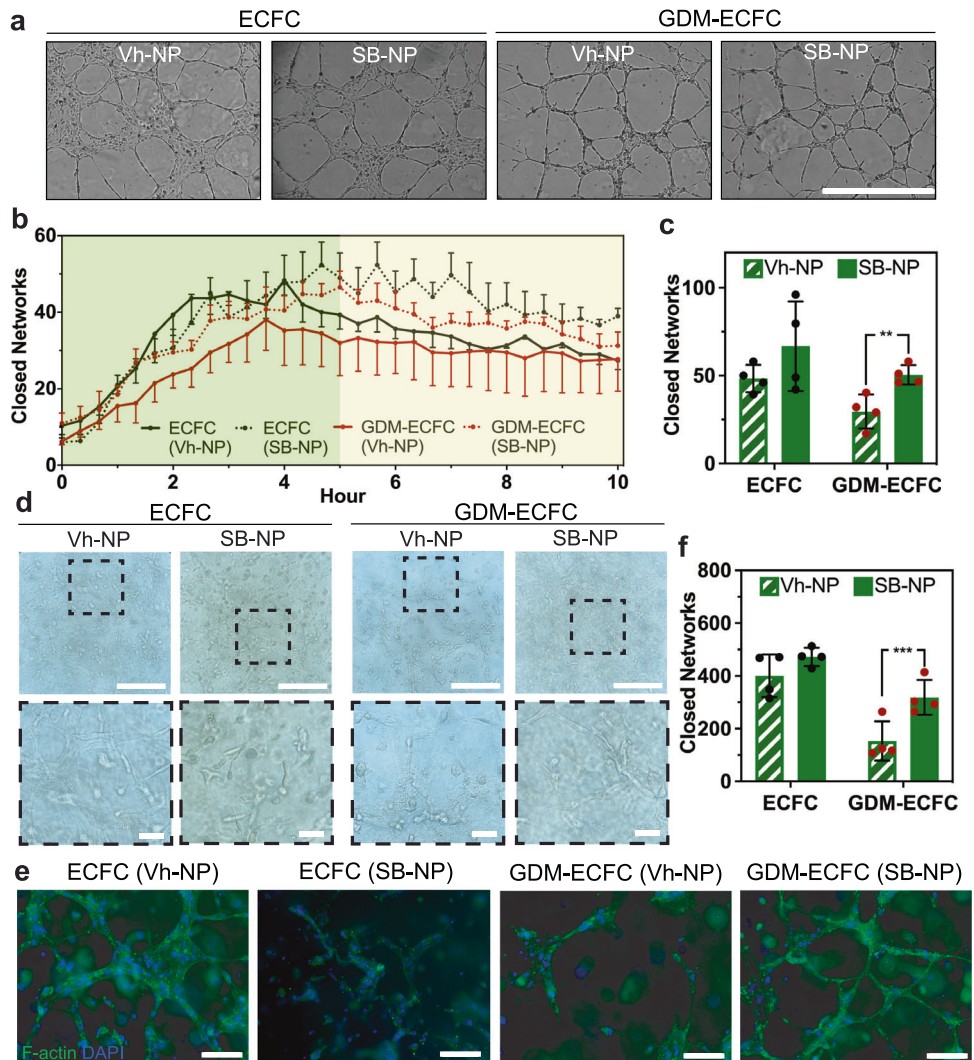

**Fig. 5 Bioactive Nanoparticles Improve in vitro Vasculogenesis. a** High contrast brightfield images of ECFCs and GDM-ECFCs vascular tube formation on Matrigel 5 h post plating following treatment with Vh-NPs or SB-NPs. Scale bar is 1 mm. **b** Kinetic analysis of vasculogenesis (KAV) identifies closed networks formed over time. **c** The number of closed networks 5 h post-plating were quantified and plotted. SB-NPs significantly improve closed networks formed by GDM-ECFCs (**$P = 0.010$), but not by normal ECFCs ($P = 0.218$). Four biological replicates ($n = 4$; mean ± s.d.) were used for normal ECFCs (*black* data dots) and GDM-ECFCs (*red* data dots). **d** High contrast brightfield images of ECFCs and GDM-ECFCs forming 3D vascular networks in collagen at 48 h post encapsulation following conjugation with Vh-NPs or SB-NPs. High magnification images of the dashed areas depict the vascular tube networks. Scale bars are 100 μm. **e** Fluorescent images of ECFCs and GDM-ECFCs forming 3D vascular networks were stained for F-actin (green) and nuclei (blue). Scale bars are 50 μm. **f** The number of closed networks 48 h post encapsulation were quantified and plotted. Four biological replicates ($n = 4$; mean ± s.d.) were used for normal ECFCs (*black* data dots) and GDM-ECFCs (*red* data dots). SB-NPs significantly improve closed networks formed by GDM-ECFCs (***$P = 0.0031$), but not by normal ECFCs ($P = 0.051$). Statistical significance was evaluated using Student's *t*-test. Significance levels were set at *$P < 0.05$, **$P < 0.01$, ***$P < 0.005$.

remained higher throughout the time course compared to GDM-ECFCs conjugated with Vh-NPs (Fig. 5b). Moreover, the difference between GDM-ECFCs conjugated with SB-NPs and Vh-NPs were statistically validated using mean kinetic values for the number of closed networks similar to our previous studies[30,31,43]. This statistical difference is within the pointwise 95% confidence intervals (error bars) and greater than zero throughout the duration of the experiment suggesting functional improvements of SB-NPs on network structure (Supplementary Fig. 8)[30,31,43]. Motivated by this promising result, we further evaluated if bioactive nanoparticles could augment vascular tube formation in a 3D collagen/fibronectin gel, which is usually performed as validation before progressing to in vivo vasculo-genesis studies[3,10,45]. First, we encapsulated ECFCs or GDM-ECFCs conjugated with either Vh-NPs or SB-NPs into 3D

collagen/fibronectin gels[3,10,45]. After 24 h of culture, 3D capillary tube formation was observed within the gels (Fig. 5d, e and Supplementary Fig. 9). Then, we utilized the KAV method to quantify the vascular closed networks formed within the 3D collagen/ fibronectin gels[31]. KAV analysis revealed that SB-NPs significantly increased 3D vascular closed networks formed by GDM-ECFCs (***$P = 0.0031$), but not by normal ECFCs (#$P = 0.051$) in comparison to Vh-NPs corresponding control groups (Fig. 5f)[31,43]. Overall, these results suggest that SB-NPs can significantly restore the 2D and 3D vasculogenic potential of GDM-ECFCs.

**Bioactive nanoparticles restore in vivo vasculogenic potential of GDM-ECFCs.** A unique characteristic of ECFCs is their ability to

form *de novo* blood vessels in vivo upon transplantation[13,46,47]. In preliminary studies, we demonstrated that GDM-ECFCs exhibit reduced functional capacity to form chimeric vessels in vivo compared to normal ECFCs (Supplementary Fig. 10a–c). Therefore, we next investigated whether SB-NPs could restore in vivo vasculogenic potential of GDM-ECFCs. Early passages (P2-5) of normal ECFCs and GDM-ECFCs conjugated with either Vh-NPs or SB-NPs were encapsulated in collagen/fibronectin gels to allow 3D capillary tube formation (Supplementary Fig. 9)[3,10,45]. After 24 h of culture, the pre-vascularized grafts were transplanted into each flank of immunodeficient NOD/SCID mice[4,45,48]. Since SB-NPs do not augment in vitro vascular formation of normal ECFCs, in vivo studies focused on assessing whether SB-NPs conjugated to GDM-ECFCs improved vasculogenic function in vivo. At 14 days, the grafts were harvested and processed for histological and immunohistochemical analysis (Supplementary Fig. 11a–d). Qualitatively, the overall size of the grafts from the GDM-ECFCs conjugated with Vh-NPs was smaller compared to the grafts from the normal ECFCs conjugated with Vh-NPs or GDM-ECFCs conjugated with SB-NPs (Fig. 6a–c). Normal ECFCs vascularized the entire graft, while GDM-ECFCs sparsely vascularized the periphery of the graft (Fig. 6a, b and Supplementary Fig. 10a–c). In contrast, GDM-ECFCs conjugated with SB-NPs vascularized the entire graft, comparable to normal ECFCs conjugated with Vh-NPs (Fig. 6a, c). Consecutive slides stained for human CD31 and H&E were used to confirm that the majority of chimeric vessels were human CD31 positive and perfused with mouse erythrocytes[49,50], suggesting graft vessels that were functionally anastomosed with host vessels (Fig. 6d, f and Supplementary Fig. 11a–d). Quantification of the chimeric vessels revealed that SB-NPs conjugation of GDM-ECFCs more than doubled vessel density compared to Vh-NPs conjugation ($44.7 \pm 11.7$ vs. $102.5 \pm 46.3$ vessels/mm$^2$, Fig. 6g). Similarly, vessel area significantly increased for GDM-ECFCs conjugated with SB-NPs compared to Vh-NPs conjugation (Fig. 6h).

To further study the dynamic interaction between the human and host vasculatures, perfusion imaging of the grafts was performed using intravital microscopy. Prior to harvesting the grafts (at day 14), mice were perfused with rhodamine-conjugated *UEA-I* lectin and fluorescein-conjugated *GS-IB4* isolectin to label the human and mouse vessels, respectively[45,51,52]. Intravital microscopy revealed that host vasculature invaded the periphery of the human vascularized grafts (Supplementary Fig. 12). Furthermore, 3D rendering of the grafts demonstrated the dynamic interaction between the human vessels stained with *UEA-I* lectin (human) and the host vasculature stained with *GS-IB4* isolectin (mouse). Chimeric vessels were detected with some overlapping stains between human- and mouse-specific lectins, which suggests that the transplanted human vascular networks were functional and anastomosing with the hosts' circulatory system (Fig. 6i). Quantification of the vessels in the grafts revealed that SB-NPs conjugation significantly increased human and mouse vessels that were interconnected to each other (Fig. 6i, j). Interestingly, SB-NPs conjugation of GDM-ECFCs also shifted the mean vessel size distribution ($10.7 \pm 3.6$ vs. $14.6 \pm 5.8$ μm, Fig. 6k). Collectively, these results suggest that SB-NPs significantly improve the in vivo vasculogenic potential of GDM-ECFCs.

## Discussion
Our study focuses primarily on fetal ECFCs isolated from infants born to mothers with GDM, which have an increased risk of developing chronic health complications, including T2DM, hypertension, and cardiovascular diseases (Fig. 7)[15,30]. In particular, the risk for cardiovascular disease endpoints was significantly elevated in offspring exposed to GDM (adjusted hazard ratio 1.42 and confidence interval 1.12–1.79)[5,53,54]. This suggests that exposure to a diabetic intrauterine environment induces premature dysfunction of ECFCs, which are present in the circulation and in vessel walls, and are highly enriched in umbilical cord blood[3,4]. A genome-wide microarray analysis conducted on cord blood-derived ECFCs identified *TAGLN* as one of the genes significantly increased in GDM-ECFCs[30]. TAGLN is an F-actin binding protein that regulates the organization of actin cytoskeletal, cellular contractility, and motility[30,32,55]. When TAGLN was first discovered, it was named as SM22α because it was considered a calponin-related protein expressed specifically in adult smooth muscle cells[56,57]. Subsequent studies demonstrated TAGLN expression in fibroblasts, epithelial cells, and multipotent mesenchymal stromal cells (MSCs), where TAGLN has a role in generating committed progenitor cells from undifferentiated MSCs by regulating cytoskeletal organization[58].

Like most healthy ECs, ECFCs express low levels of TAGLN[4,9,30]. However, exposure to high glucose and inflammatory cytokines induces TAGLN expression through TGF-β and IL1-β signaling pathways[30,55]. Increased TAGLN expression does not represent trans differentiation of GDM-ECFCs to other cell types, rather it suggests a transition of GDM-ECFCs into an unstable state, which can be reversed with the administration of exogenous therapeutics. In fact, knocking down TAGLN expression with siRNA rescues cell migration and vasculogenic potential of GDM-ECFCs[30]. Collectively, these observations provided rationale for evaluating whether a TGF-β inhibitor could enhance the vasculogenic function of GDM-ECFCs.

Inspired by pioneering studies that conjugated drug-cargos to the surface of HSPCs[38,59,60], we engineered bioactive NPs that can directly attach to the cell surface and deliver therapeutic agents to the GDM-ECFCs. The multilamellar structure of the NPs allows high encapsulation efficiency of the therapeutic reagents (above 70%), long-term stability, and sustained release via diffusion up to 14 days, which matches the timeline required for ECFCs to undergo vascular anastomosis in vivo[45,48,51]. Similar to other circulating progenitor cells[37,38], we demonstrated that ECFCs express high level of free thiols on the cell surface, which can be used for benign thiol-maleimide coupling. Following NPs conjugation to the cell surface, the residual reactive groups on the NPs were quenched using a PEGylation strategy[61,62], which has been widely used to improve NPs-based drug delivery and to minimize the immunogenicity of NPs[63,64]. Moreover, this PEGylation strategy is crucial to avoid non-specific binding of these NPs to the surface of other cells (Supplementary Figure 5). Since the size of a typical ECFC is larger than HSPC, it is likely that there are more free thiols available on the surface of ECFC than HSPC[38,65]. We demonstrated that we can conjugate up to 5000 NPs per cell, while preserving cellular function and progenitor phenotypes of ECFCs. We did not attempt to conjugate more than 5000 NPs per cell because beyond this cells to NPs ratio the fluorescence signal starts to deviate from linearity (Fig. 1f), which may suggest that NPs can start to fuse together due the increased steric packing of NPs on the surface of the cell membrane[22,36]. Indeed, at this optimum ratio the engineered NPs provide a sustained and controlled release of SB-431542, which acts as an inhibitor of the TGF-β/SMAD3 pathway. Since SB-431542 is an amphiphilic molecule with both polar and hydrophobic groups, it can interact with the lipid membrane of the liposomes[66], which in turn contributes to its slower release at higher loading concentrations. By competing for the ATP binding site, SB-431542 selectively inhibits the TGF-β type I receptor ALK5, whereas it does not affect the BMP type 1 receptors like ALK2, ALK3, ALK6[67]. Selective targeting of the ALK5/SMAD3 pathway can directly downregulate TAGLN expression [32].

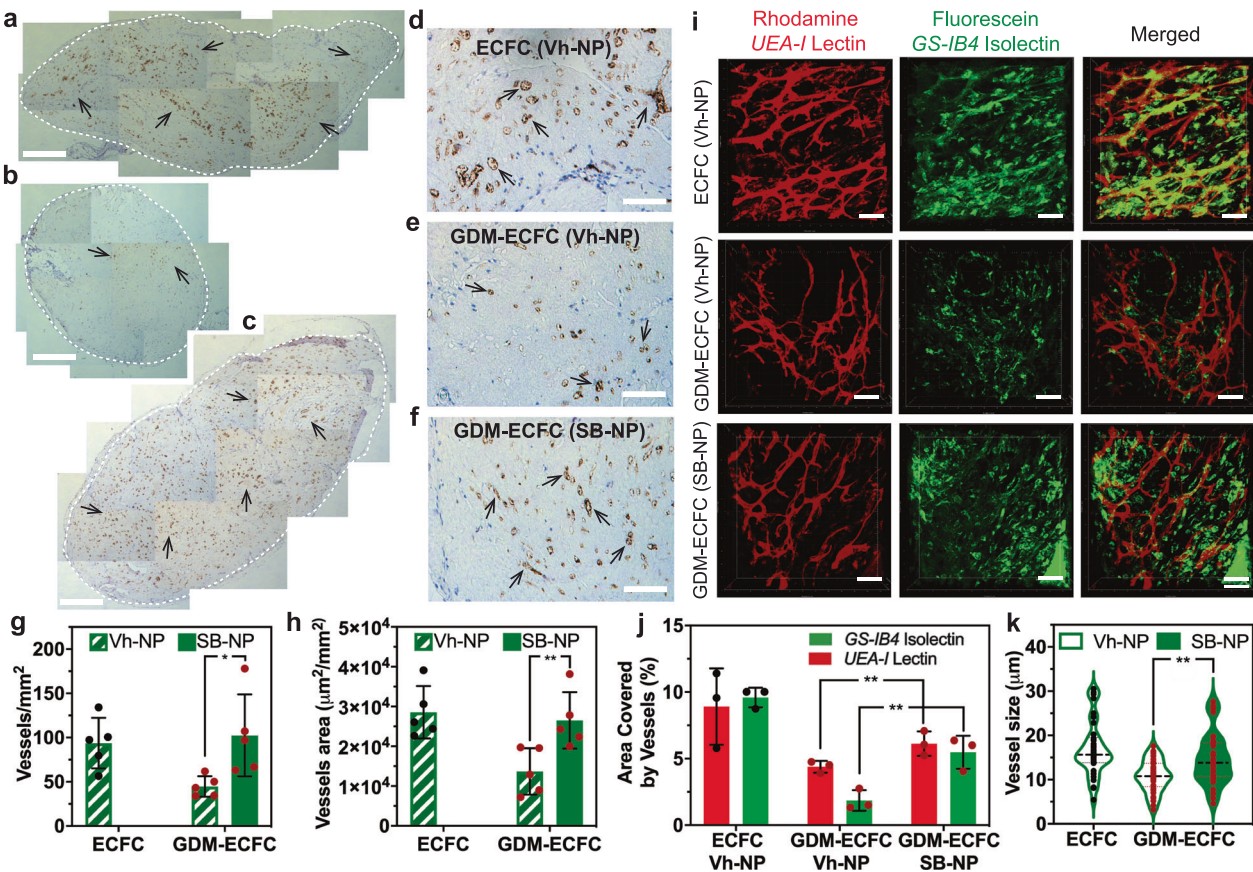

**Fig. 6 In vivo functionality of ECFCs conjugated with bioactive nanoparticles.** Normal and GDM-ECFCs were conjugated with nanoparticles containing either vehicle (Vh) or 40 μM SB-431542 (SB). Conjugated ECFCs (100,000 cells/gel) were encapsulated in collagen/fibronectin gels for 48 hr to form vascularized networks and then transplanted into the side flanks of NOD/SCID mice (*n* = 4-6 animals/group). Representative IHC images stained with anti-human CD31 illustrate the graft harvested 14 days following transplantation (highlighted in white dotted circle) for (**a**) ECFCs (Vh-NP), (**b**) GDM-ECFCs (Vh-NP), and (**c**) GDM-ECFCs (SB-NP). Scale bars are 500 μm. Representative high magnification IHC images of grafts from (**d**) ECFCs (Vh-NP), **e** GDM-ECFCs (Vh-NP), and (**f**) GDM-ECFCs (SB-NP) after 14 days implantation into NOD/SCID mice stained with anti-human CD31 (brown). Arrows indicate human CD31+ vessels, which are perfused with murine erythrocytes. Scale bars are 30 μm. The number and size of human CD31+ vessels were quantified and plotted. Compared to the Vh-NPs control, SB-NPs conjugation increases (**g**) vessel density (*$P$ = 0.011) and (**h**) vessel area (**$P$ = 0.0058) formed by GDM-ECFCs in vivo. Five animals (*n* = 5; mean ± s.d.) were used to evaluate each group: normal ECFCs (*black* data dots) and GDM-ECFCs (*red* data dots). **i** Representative intravital images of grafts pre-perfused with rhodamine-conjugated *UEA-I* lectin to stain the human vessels (in red) and fluorescein-conjugated *GS-IB4* to stain the mouse vessels (in green). 3D confocal rendering demonstrated the interaction between human vasculature (in red) and mouse vasculature (in green) for normal ECFC, GDM-ECFCs conjugated with Vh-NPs and GDM-ECFCs conjugated with SB-NPs. Scale bars are 40 μm. **j** Vessel quantification and analysis reveal the percent area covered by human vasculature (*UEA-I* lectin) and mouse vasculature (*GS-IB4* isolectin). SB-NPs conjugation to GDM-ECFCs results in the significant increase of both human vessels (**$P$ = 0.021) and mouse vessels (**$P$ = 0.006) that were interconnected to each other. Three animals (*n* = 3; mean ± s.d.) were used to evaluate each group: normal ECFCs (*black* data dots) and GDM-ECFCs (*red* data dots). **k** Vessel quantification reveals the size distribution of the vessels found in the explant for normal ECFCs, GDM-ECFCs conjugated with Vh-NPs and GDM-ECFCs conjugated with SB-NPs. Compared to the Vh-NPs control, SB-NPs conjugation results in an increase in the mean of vessel size distribution from for GDM-ECFCs (**$P$ = 0.0018).

We have previously utilized siRNA to downregulate TAGLN expression in GDM-ECFCs, which resulted in the improvement of cell migration and vascular tube formation in vitro[30]. In this study, we tested if SB-NPs conjugated to the surface of GDM-ECFCs can improve the clinical potential of GDM-ECFCs, without the anticipated vascular and immune effects associated with siRNA therapeutics[29]. To evaluate our regenerative strategy, we conducted a 3D vasculogenesis assay, where ECs have been shown to form functional vascular networks with lumen structures within collagen/fibronectin gels[10,68,69]. Using KAV analysis to quantify the vascular networks in 3D collagen/fibronectin constructs[31,43], we demonstrated a significant increase in the closed networks formed by GDM-ECFCs conjugated with SB-NPs compared to GDM-ECFCs conjugated with Vh-NPs control. It is interesting to note that SB-NPs conjugated to normal ECFCs does not significantly affect cell migration or vascular tube formation, which may be because these cells express normal TAGLN levels[15,30]. It is possible that the improved functional effects observed in GDM-ECFCs with SB-NPs were not entirely due to changes in TAGLN expression, since TGF-β inhibition may effect expression of other proteins in ECFCs[70]. Nonetheless, these data are encouraging and support the notion that specifically targeting disrupted molecular pathways may reduce off-target effects of normal cells in vivo[71]. Moreover, given that improvements in the functional properties of GDM-ECFCs is associated with reduced TAGLN expression, TAGLN could serve as a molecular marker to screen for future small molecule compounds.

Since the hallmark of ECFCs is their ability to form *de novo* blood vessel formation in vivo, our next investigations focused on determining if SB-NPs can restore in vivo vasculogenic potential

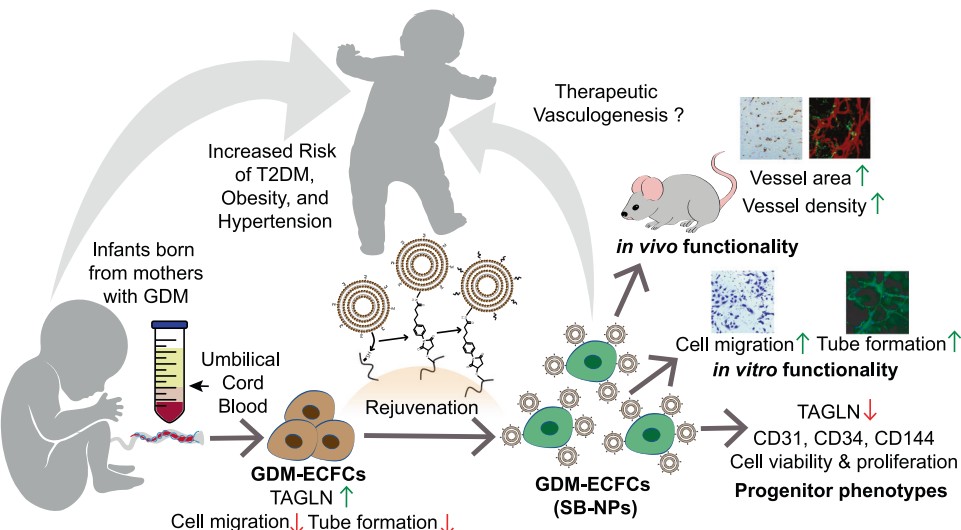

**Fig. 7 Schematic representation of the cell surface conjugation strategy and functionality study.** Children born from mothers with GDM experience increased risk of developing T2DM, hypertension, and cardiovascular disease later in life. Key to this enhanced risk is stress-induced dysfunction of vascular progenitor cells, including ECFCs. Cord blood-derived ECFCs isolated from patients with GDM overexpress TAGLN, demonstrate reduced cell migration, and exhibit impaired tube formation compared to normal ECFCs. Cell surface conjugation with bioactive nanoparticles enable pseudo-autocrine stimulation of GDM-ECFCs with SB-431542. Rejuvenation of GDM-ECFCs normalizes TAGLN expression, preserves key progenitor phenotypes, improves cell migration and tube formation in vitro, as well as augments the density and area of chimeric vessels formed in vivo. This cell surface conjugation strategy to rejuvenate GDM-ECFCs and improve their clinical potentials can be used for therapeutic vasculogenesis to address various complications precipitated from GDM.

of GDM-ECFCs[13,47,72]. The subcutaneous xenograft model employed in this study creates a consistent and robust vasculogenic response by ECFCs[13,46,49,52,73,74]. As such, this model is the preferred small animal study to compare vasculogenic potential among ECFCs from different cell sources or experimental treatments[3,15,75,76]. Hence, based on previous studies that analyzed the kinetics of human and host vessels integration[45,48,51,77], the xenografts were analyzed using intravital microscopy and immunohistochemistry after 14 days of implantation. We observed that GDM-ECFCs were only able to sparsely vascularize the periphery of the grafts, which may explain why the grafts containing GDM-ECFCs were relatively smaller in size compared to grafts containing normal ECFCs. Interestingly, SB-NPs conjugation significantly increased the vessel density and vessel areas formed by GDM-ECFCs in vivo, as well as restored their vasculogenesis potential to a comparable level to normal ECFCs[10,78]. Using consecutive slides stained for human CD31 and H&E, we observed that the majority of the chimeric vessels formed within the grafts were stained positive for human CD31 and perfused with mouse erythrocytes, which suggests functional vessels that were anastomosed with the host vessels[10,74,78]. Intravital microscope analysis was also used to confirm the functionality of the human vessels and study the interactions between the human and mouse vessels[50,51]. Further quantification of the percent area covered by vessels in the grafts revealed that SB-NPs conjugation resulted in a significant increase of both human vessels and mouse vessels that were interconnected to each other. Since it was previously shown that the host vasculature can connect with the human vasculature through the "wrapping and tapping" mechanism[51], the increase in both human and host vessels in the grafts indicate functional chimeric vascular networks that were able to integrate with the host vasculatures. Altogether, these results suggest that SB-NPs can significantly improve the in vivo vasculogenic potential of GDM-ECFCs.

We report a simple, yet promising strategy to conjugate bioactive NPs directly onto the surface of ECFCs to improve their vasculogenic potential (Fig. 7)[23,79]. We envision that this platform technology can be used in the setting of current clinical practice, where stem and progenitor cells are regularly being isolated and processed ex vivo prior to direct stem cell transplantation or stem cell banking for future use[13,80,81]. Since the rejuvenation strategy seems to benefit the most for GDM-ECFCs with relatively high TAGLN expression, it is logical to use TAGLN as biomarker to define the threshold point where the clinical benefits could potentially outweigh the cost and risk of the therapeutic approach. Promising results from this study warrant future investigations to systematically define the selection criteria for suitable candidates in clinical trials. While this approach is not intended to confer a lifetime of protection against elevated risk of cardiovascular risk, the rejuvenated stem and progenitor cells can be used in the events where direct stem cell transplantation or engineered tissues are needed to address cardiovascular complications, such as peripheral artery disease and diabetic non-healing wounds[79,82,83]. Moreover, such a simple cell-surface engineering strategy can be broadly applied to improve dysfunctional ECFCs from the peripheral blood of patients with chronic diseases (i.e., T2DM)[3,17] and from umbilical cord blood of patients with complicated pregnancies (i.e., preeclampsia)[75,76,84], as well as to improve the expansion and maintenance of vascular progenitor cells derived from hPSCs[18,40–42]. This platform technology also has the potential to increase autologous donor cell pools by enhancing the functional capacity of progenitor cells previously considered unsuitable for cell therapy[85,86]. Collectively, our study demonstrates a promising platform to rejuvenate vascular progenitor cells, which can be clinically-translated to improve non-invasive, cellular based treatments for cardiovascular complications and to enhance current approaches for tissue repair and regenerative medicine[12,26].

## Materials and methods

**Umbilical cord blood sample acquisition**. Human umbilical cord blood samples (40–60 mL) were collected in heparinized solution at the time of birth for normal / uncomplicated and GDM pregnancies (gestational age 38-42 weeks) following written informed consent[4,15]. GDM was defined per American College of

Obstetrics and Gynecology guidelines. Exclusion criteria include T1DM or T2DM, illness known to affect glucose metabolism (i.e., Cushing syndrome, polycystic ovarian syndrome), use of medications that affect glucose metabolism (i.e., dexamethasone), multiple gestation, history of pre-eclampsia, cardiovascular disease, and women carrying fetuses with chromosomal abnormalities[15,87]. Historical and clinical data were obtained at each visit and at the time of delivery (Supplementary Table 1). Maternal blood was collected for glycosylated hemoglobin (HgA1C) and oral glucose tolerance test (GTT). Cord blood was collected and processed in our AngioBioCore facility for human mononuclear cells (MNCs) used for ECFC isolation and assays. The Institutional Review Board at the Indiana University School of Medicine (IUSM) approved all protocols, and informed consent was obtained from all women.

**Isolation and characterization of ECFCs.** The human ECFCs were isolated and characterized as previously described[3,10]. Briefly, tissue culture plates pre-coated with collagen I were seeded with human MNCs in complete endothelial growth medium-2 (EGM-2). After 24 h of culture, nonadherent cells were aspirated and complete EGM-2 medium was added to each well. Colonies of endothelial cells appeared between 5 and 8 days and were identified as monolayers of cobblestone-appearing cells. ECFCs were characterized for the positive expression of cell-surface antigens CD31, CD141, CD105, CD144, vWF, and Flk-1, as well as negative expression of hematopoietic-cell surface antigens CD41 and CD14. Single cell colony forming assays were used to characterize their robust and proliferative potential, secondary and tertiary colony formation upon plating. Normal or uncomplicated ECFC lines include E1-CB-111, E1-CB-150, E1-CB-153, E1-CB-157; GDM-ECFC lines include E1-CB-36, E1-CB-37, E1-CB-71, and E1-CB-74 (Supplementary Table. 1). To maintain the ECFC culture, flasks or well plates were pre-coated with rat-tail collagen type I solution (50 μg/mL, Corning), then incubated for at least 3 h at 37 °C and washed with PBS three times before use. Complete medium for growing ECFCs consisted of EGM-2 (Promocell, C-22011), supplement Mix (Promocell, C-39216), and 0.2% mycoZapTM Prophylactic (Lonza). The cells were maintained at 37 °C, 5% $CO_2$, passaged using DetachKit (Promocell, C-41222), and used for experiments between passages 2-5. All cell lines were routinely tested for mycoplasma contamination and were negative throughout this study.

**Nanoparticle fabrication.** Synthesis of multilamellar liposomal nanoparticles (NPs) was performed based on the standard thin film lipid hydration method[35,38]. The components consist of phospholipids (Avanti® Polar Lipids) including MPB-PE (1,2-dioleoyl-sn-glycero-3-phosphoethanolamine-N-[4-(p-maleimidophenyl)butyramide] (sodium salt)), DOPC (1,2-dioleoyl-sn-glycero-3-phosphocholine), DOPG (1,2-dioleoyl-sn-glycero-3-phospho-(1'-rac-glycerol) (sodium salt), fluorescent tracker 1,1'-Dioctadecyl-3,3,3',3'-tetramethylindocarbocyanine perchlorate (Dil, Sigma-Aldrich), and SB-431542 (SB, Stemcell Technology) (Supplementary Fig. 1a). DOPC, DOPG, MBP-PE, SB-43154 and 'Dil' dye lipid components were combined in a glass vial and vacuum dried (−25 mm of Hg, 21 °C) for 30–45 min to obtain the dry lipid film. To fabricate liposomal nanoparticles, the dry lipid film containing DOPC/DOPG/MPB-PE/Dil/SB (1188/303/1890/120/3.6 μg) was hydrated with 1 ml PBS solution. The resulting mixture was vortexed for 5–7 min, then extruded 21 times using gas tight syringes, through a 200 nm polycarbonate membrane sandwiched in the Mini-Extruder block (Avanti® Polar Lipids) to improve monodispersity of small liposomal vesicles (<200 nm). The resulting solution was incubated at room temperature for 1.5 h, then at 4 °C overnight. The NPs were purified from free phospholipids and free SB by ultracentrifugation at 50,100 rpm at 4 °C for 2.5 h (ThermoScientific, Sorvall MX120 + Micro-Ultracentrifuge, Rotor S55-A2). The SB-NPs were obtained by resuspending the pellet in PBS.

**Conjugation of drug-loaded nanoparticles to ECFCs.** Drug-loaded nanoparticles were conjugated on the surface of the cells by mixing equal volumes of ECFCs and SB-NPs in nuclease-free water, with nanoparticles to cell ratios ranging from 1:100 to 1:5,000[22,35]. The cells were then incubated for 30 min at 37 °C with gentle agitation to facilitate the conjugation of maleimide in liposomes to the free thiols on ECFCs. The residual maleimide groups on cell-bound particles were quenched with 1 mg/mL thiol-terminated 2-kDa PEG for 30 min in complete EGM-2 medium. After nanoparticles were conjugated on the cell surface, their presence was confirmed by confocal microscopy (Nikon A1R-MP). Mean fluorescence intensity (MFI) was measured using FACS (BD LSR Fortessa X-20) and FlowJo software.

**In vitro functional assays.** Human ECFCs and GDM-ECFCs at passage 2–5 were assessed for their in vitro function using cell migration, proliferation, wound healing, and vasculogenesis assays as previously described[44,88,89]. FACS, western blot, qRT-PCR were used to analyze protein and gene expression levels. Details methods with information on primers and antibodies are available in the Supplementary Information and Supplementary Table 3. Tube formation was quantified using the KAV plugin in the FIJI program[31,90].

**In vivo functional assays.** Subcutaneous implantation of the cell-containing gel plugs was conducted with 8-week old NOD-SCID mice following the procedure approved by Indiana University School of Medicine IACUC[72,73]. Briefly, mice were anesthetized by isoflurane. A small incision was made to create two subcutaneous pockets near the dorsal flanks. One gel plug was inserted into each pocket, one with Vh-NPs and one with SB-NPs. The incision was clipped and Ketoprofen (100 mg/ml) was injected for reducing pain. A perfusion study was performed on day 14 following the transplantation. Rhodamine-conjugated *Ulex Europaeus Agglutinin I* (*UEA-I* lectin, Vector Laboratory, dilution ratio 1:2) and fluorescein-conjugated *isolectin Griffonia simplicifolia* (*GS-IB4* isolectin, Vector Laboratory, dilution ratio 1:2) were injected through retro-orbital veins[45,51]. After 20 min, intravital images were taken using a dissecting microscope (Leica, magnification 12X) and multi-photon microscope (Olympus). Confocal image stacks were acquired to create 3D rendering images, which were quantified for percent area covered by human and mouse vessels, as well as for distribution of vessel diameters using ImageJ. On day 14, a separate group of mice were euthanized and the gel plugs were harvested and fixed with 4% paraformaldehyde. Tissue samples were processed, sectioned and stained for H&E at the Histology and Histomorphometry Core, Indiana Center for Musculoskeletal Health, IUSM. Sectioned slides were then stained for H&E and IHC using human CD31 (clone JC70A, Dako), mouse CD31 (SC1506, Santa Cruz), and mouse SMA (SC53015, Santa Cruz), as well as appropriate IgG isotype controls (Supplementary Fig. 11)[4,10]. The number of human blood vessels and size were counted, measured, and normalized to the graft area. We sampled a minimum of 10 images for each graft, analyzed, and normalized the number and size of blood vessels accordingly. Consecutive slides stained for human CD31 and H&E were used to confirm hCD31+ vessels with perfused mouse erythrocyte (Supplementary Fig. 10). Functional vessels within the explants were counted only if they contained at least 1 mouse erythrocyte[18].

**Statistics and reproducibility.** All experiments were performed with at least four biological replicates ($n = 4$) conducted in triplicate. Data are presented as mean ± standard deviation, unless otherwise were specified in the figure legends. A power analysis with a 95% confidence interval was used to calculate sample size required to obtain statistically significant results. The sampling number we used gave a normal distribution. All statistical analysis were conducted in GraphPad Prism. Statistical comparisons were made using Student's $t$ test for paired data, analysis of variance (ANOVA) for multiple comparisons, and with Tukey post hoc analysis for parametric data. Significance levels were set at the following: *$P < 0.05$, **$P < 0.01$, ***$P < 0.001$, ****$P < 0.0001$.

**Reporting summary.** Further information on experimental design is available in the Nature Research Reporting Summary linked to this paper.

## Data availability
Additional data which contributed to this study are present in the Supplementary Information and source data can be found in Supplementary Data 1.

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

## Acknowledgements

We acknowledge support from the University of Notre Dame through "Advancing Our Vision" Initiative in Stem Cell Research and Scientific Wellness Initiative, Harper Cancer Research Institute – American Cancer Society Institutional Research Grant (IRG-17-182-04), American Heart Association through Career Development Award (19-CDA-34630012 to D.H.-P.), and from National Institutes of Health (R01-HL-094725 to L.S.H. and R35-GM-143055 to D.H.-P.). We would like to thank the Notre Dame Integrated Imaging Facility for performing TEM imaging and the Immunohistochemistry Core Facility at the Indiana University School of Medicine for performing the IHC staining and analysis, as well as AngioBioCore at Indiana University Simon Comprehensive Cancer Center (NIDDK/NIH U54-DK-106846 and P30 CA-082709) for isolating and characterizing cord blood samples. This publication was made possible, with support from the Indiana Clinical and Translational Science Institute (I-CTSI) funded, in part by Grant Number ULITR001108 from the NIH for Advancing Translational Sciences, Clinical and Translational Science Awards.

## Author contributions

L.B., S.E., L.S.H., and D.H.-P. conceived the ideas, designed the experiments, interpreted the data, and wrote the manuscript. L.B., S.E., E.H., L.A., K.R., M.O., P.S., and P.D.N. conducted the experiments and analyzed the data. S.Z. and L.B. conducted the perfusion study and intravital imaging. L.S.H. and D.H.-P. supervised the study. All authors have approved the manuscript.

## Competing interests

The authors declare no competing interests.

## Additional information

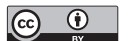

