## [Peer Review file · Communications Biology]

Engineering Bioactive Nanoparticles to Rejuvenate Vascular Progenitor CellsReviewers' comments:

Reviewer #1 (Remarks to the Author):

Thank you for the opportunity to review this manuscript by But et al. The manuscript describes feasibility experiments to bind nanoparticles (NP) and NP-loaded with inhibitor to bind to ECFC and potentially alter function of ECs in vitro and in vivo. The author's demonstrate their findings in a logical approach that was easy to follow. Overall, I am enthusiastic about their methodology and the potential for this approach to alter EC expression and function. I do have a couple of suggestions to improve the manuscript findings and clarity of the results.

Major Questions:

1. The authors clearly show that NP bind ECFC and release drug. Have the authors considered whether NP binding to EC is reversible under some conditions and that NPs may release from EC and bind to other EC. For example, if you take culture media from NP-EC and place on control (no NP) EC, do the authors see NP binding? This would also be interesting to know whether the inhibitor could work on unbound EC through some similar releasing mechanism. Either way, I think this experiment would support specificity and Figure 1 & 2 results or open some alternative thoughts about NP-loaded with inhibitors.
2. While the authors rely on the lack of immunogenicity of NP from other studies in HSCs, it would be helpful (and complete) to demonstrate similar findings for NP created in this study and NP-labeled EC.
3. Have the authors considered providing NP into umbilical cords themselves to see if they bind EC under more clinically-relevant conditions?

Minor Questions:

1. Could the authors comment on the quantity of thiol groups in EC vs HSC?
2. NP binding does not appear to be linear in Fig 1F. Do the authors believe they bind to themselves or fuse?
3. Would be helpful to quantify functional vessels in the collagen gel since this reviewer cannot appreciate mouse RBC in human vessels.
4. Figure 3 layout is not logical. Perhaps placed A-C on one row with D and E below would be more logical? Also, should the authors draw lines between paired samples in Figure 3E since the authors state that there is a connection between TAGLN expression and degree of GDM (presumed GTT)? Demonstrating pairs would also be helpful since some GDM-ECFC express similar TAGLN under both control and treatment conditions (e.g Lanes 1 and 2; Lanes 4 and 5).
5. Please use angiogenesis or vasculogenesis but not both (e.g. Figure 5)
6. Line 182 reads "for at least after 6 days in culture". Please correct.

Reviewer #2 (Remarks to the Author):

This manuscript by Bui et al. presents the studies on the application of small molecule SB-431542 loaded liposomal nanoparticles directly conjugated to the surface of GDM-exposed ECFCs to rejuvenate these circulating vascular progenitor cells. Their results supported the conclusion that controlled delivery of SB-431542 to GDM-ECFCs normalized transgelin (TAGLN) expression and

improved cell migration, and eventually enhanced in vitro and in vivo vasculogenesis of GDM-ECFCs. Overall, the study presents a novel approach for improving vasculogenic potential of GDM-exposed ECFCs using TGF- β inhibitor that can potentially overcome the challenges associated with growth factor delivery or gene therapy. The study is carefully developed, and the manuscript is well-organized. However, some revisions and clarifications are required to strengthen the data and improve the quality of the manuscript, which are listed below:

1. The attachment of nanoparticles on the surface of ECFCs as well as cell viability was studied using nanoparticles without SB incorporation. Considering the hydrophobic nature of small molecule SB-431542, the authors are suggested to perform studies to investigate the nanoparticles conjugation and cell viability using SB incorporated nanoparticles.
2. The proposed drug delivery approach for rejuvenating GDM-ECFCs is based on the conjugation of the lipid particles to free thiols on the surface of GDM-ECFCs. How the conjugation of these nanoparticles to the thiol groups on the surface of other cell types including mononuclear cells or other vascular cells might adversely affect the application of SB-NPs for the intended application?
3. In order to study the efficacy of SB-NPs in vivo, GDM-ECFCs were pre-conjugated with SB-NPs and encapsulated in the gels and then transplanted subcutaneously. How the authors would justify the application of their developed drug delivery system for clinical application? What is the chance of SB-NPs being up taken by circulating GDM-ECFCs if the nanoparticles are administered locally or systemically?
4. Please provide the raw data and detailed statistical analysis for the figures listed below as supplemental data:

Figure 3B and 3E

Figure 4B and 4E

Figure 5C and 5F

Figure 6G and 6H

5. The authors are recommended to remove the “#” symbol used to show the statistical insignificance and only use the symbols where the comparison is statistically significant, to eliminate the chance of any confusion.
6. What kind of CD31 mouse specific antibody was used in the study? Please specify it in the methods section.
7. Please explain the in vivo perfusion procedure in the respective methods section.

Reviewer #3 (Remarks to the Author):

The authors study the ability of drug-loaded nanoparticles to restore the activity of endothelial colony forming cells derived from infants born to mothers with gestational diabetes. These cells may be useful for therapeutic angiogenesis to prevent development of cardiovascular diseases later in life. The study is based on the authors' previous work demonstrating attenuation of elevated TAGLN expression can restore function of endothelial cells derived from this population. In this study, nanoparticles are bound to the cell surface to deliver a signaling molecule known to inhibit TAGLN expression. The authors demonstrate improved migration and vasculogenesis using in vitro and in vivo assays. Use of blood samples from infants born to mothers with gestational diabetes is a strength of this study due to the clinical relevance. While the data convincingly demonstrate improved functionality of the endothelial cells, their therapeutic benefit for preventing diseases that manifest over the course of

decades is uncertain.

1. There are several typos that need to be corrected. Line 35 should read "restore endothelial function." Line 137 should read "without affecting." Line 327 should read "without the anticipated vascular and immune effects." Line 362 should read "the increase in both human." Line 363 should read "indicate functional chimeric vascular networks."
2. The authors should provide statistics to demonstrate the significance of the clinical problem. What is the incidence of gestational diabetes? What is the incidence of cardiovascular disease for individuals exposed to gestational diabetes versus the general population?
3. How do the authors envision their nanoparticle-conjugated cell therapeutic will be used clinically? Would this be a one-time treatment or a recurring treatment? Would the cells be injected intravenously? When and how often? Is improved angiogenesis at 2 weeks post treatment expected to confer a lifetime of protection against elevated risk of cardiovascular disease?
4. Would all babies born to mothers with gestational diabetes be candidates to receive and potentially benefit from this treatment? Or would there be a screening and selection process to identify suitable candidates? If so, based on what criteria?
5. The authors state 5,000 particles per cell is the maximum that could be conjugated without affecting proliferation and viability. However, it does not appear that higher doses were studied. How can the authors conclude this is the maximum?
6. Lines 183-184 state protein expression correlates well with gene expression. This statement is too strong as protein expression did not decrease to nearly the same extent as gene expression (Fig. 3C vs. 3E).
7. How were mouse erythrocytes identified in chimeric vessels? Based on morphology?
8. In Figure 1H, why does higher drug concentration result in reduced release rate?
9. Figure 1C and 1D, please add units to the vertical axes.

Response to Reviewers

Reviewer #1:

Thank you for the opportunity to review this manuscript by But et al. The manuscript describes feasibility experiments to bind nanoparticles (NP) and NP-loaded with inhibitor to bind to ECFC and potentially alter function of ECs in vitro and in vivo. The author's demonstrate their findings in a logical approach that was easy to follow. Overall, I am enthusiastic about their methodology and the potential for this approach to alter EC expression and function. I do have a couple of suggestions to improve the manuscript findings and clarity of the results.

We thank the reviewer for the critical reading and thoughtful critique of our manuscript. We are very excited that the reviewer found our study interesting and can benefit the scientific community. We have attempted to address all comments and concerns as described in detail below. We believe that the manuscript is greatly enhanced as a result of the constructive feedback from the reviewers.

Major Questions:

1. The authors clearly show that NP bind ECFC and release drug. Have the authors considered whether NP binding to EC is reversible under some conditions and that NPs may release from EC and bind to other EC. For example, if you take culture media from NP-EC and place on control (no NP) EC, do the authors see NP binding? This would also be interesting to know whether the inhibitor could work on unbound EC through some similar releasing mechanism. Either way, I think this experiment would support specificity and Figure 1 & 2 results or open some alternative thoughts about NP-loaded with inhibitors.

The reviewer raised a very interesting point here. We have addressed this with the following experiment, as suggested by the reviewer. Following NPs conjugation, ECFC-NPs were cultured on tissue culture plate for 24 hours. Then, the culture media was transferred into control ECFCs (without NPs). After 24 hours of culture, the supernatant-treated ECFCs and ECFC-NPs were analyzed using FACS and confocal microscopy. FACS analysis revealed that the supernatant-treated ECFCs express a background level of the Dil dyes compared to ECFC-NPs (**Fig R1A**). Similarly, confocal analysis did not show any Dil-labeled NPs on the supernatant-treated ECFCs (**Fig R1B**). Collectively, these results suggest that once conjugated onto the surface of the ECFCs, there is a low probability for the PEGylated-NPs to be released by the ECFCs and able to bind to other ECFCs. These results agree with previous studies that showed following PEGylation the maleimide groups on the surface of the NPs are no longer reactive and therefore cannot bind to the free thiol groups on the surface of other cells. We have included these results as part of **Supplementary Figure 5** and discussed their implications in the revised manuscript.

2. While the authors rely on the lack of immunogenicity of NP from other studies in HSCs, it would be helpful (and complete) to demonstrate similar findings for NP created in this study and NP-labeled EC.

We thank the reviewer for highlighting this critical point. We envision to use this NPs technology to enhance the functionality of ECFCs in the context of an autologous stem cell transplantation.^{1,2} Toward that goal, in this study we utilized PEGylated NPs which have been widely used for clinical applications due to their lack of immunogenicity.^{3,4} While investigating the immunogenicity of ECFC-NPs is important for its translational potential, we would argue that studying this in the current research setting is neither experimentally doable nor clinically applicable. Since we utilized human ECFCs conjugated with NPs, the *in vivo* functionality studies were done in immunocompromised NOD/SCID mice, which are not suitable for studying immunogenicity.⁵ In order to carefully study the immunogenicity of our technology, we would have to isolate ECFCs from mice, conjugate them with NPs, and transplant them into immunocompetent mice.⁶ However, circulating ECFCs are very rare in mouse and very difficult to isolate from mouse peripheral or cord blood.^{7,8} Even if we are able to perform this challenging experiment, the immunogenic response could potentially be very different in human. Therefore, we believe that repeating the immunogenicity study of the NPs in the context our of current study will unlikely yield a meaningful outcome to further strengthen our study.

3. Have the authors considered providing NP into umbilical cords themselves to see if they bind EC under more clinically-relevant conditions?

We thank the reviewer for raising this intriguing question. Most clinical trials using cord blood stem cells require downstream processing to isolate a pure population of stem cells (i.e., HSCs, ECFCs).^{1,9} These stem and progenitor cells are then being directly transplanted into patients or banked for future use. While the reviewer raised a very intriguing question, the authors don't believe that testing the binding of NPs to umbilical cords would be relevant in the setting of current clinical practices. Since the unquenched maleimide groups in the NPs will bind to any cells expressing free thiol groups on their

surfaces, the authors speculate that they will bind to endothelial cells, as well as other circulating blood cells in the umbilical cords. We have included this discussion in the revised manuscript to clarify this interesting point raised by the reviewer.

Minor Questions:

1. Could the authors comment on the quantity of thiol groups in EC vs HSC?

Previous studies reported substantial amounts of free thiols on the surface of T cells (CD3⁺), B cells (B220⁺), and HSPCs (c-Kit⁺), but low amounts on RBCs (Ter-119⁺).^{3,4} Since ECFCs are larger than HSPCs, we are able to fit 5-times more NPs (~150 nm in diameter) onto the surface of ECFCs than HSPCs. Accordingly, we suspect that the quantity of free thiol groups in ECFCs is 5-times higher than in HSPCs. We have included this discussion in the revised manuscript.

2. NP binding does not appear to be linear in Fig 1F. Do the authors believe they bind to themselves or fuse?

The NPs cannot bind to themselves as they do not have any thiol groups that would react with any residual maleimide. The data from Fig.1C also indicates how stable the NPs are at 37 deg C. The MFI from samples with Cells: NPs at the ratio of 1:2500 and 1:5000 is when the signal deviates from linearity (**Fig 1F**). Considering the NPs are ~0.15 microns in diameter, there is a possibility that at cells-NPs ratios of 1:2500 and 1:5000 there might be the fusion of NPs on the cell surface, after conjugation to the cell membrane, due to diffusion and increased steric packing of the NPs on the cell membrane. This might contribute to the deviation from linearity at cells-NPs ratios of 1:2500 and 1:5000.

3. Would be helpful to quantify functional vessels in the collagen gel since this reviewer cannot appreciate mouse RBC in human vessels.

We thank the reviewer for highlighting this important point. Since the human CD31 antibody (JC70A clone) stained very strongly the vessel walls, sometimes it can be quite hard to visualize the RBC inside the vessels as pointed by the reviewer. Therefore, the authors would like to clarify that we also used H&E slides that was sectioned consecutively to follow the human CD31 slides. For instance, here we demonstrated some representative images of vessels with various shapes and sizes that were analyzed for human CD31⁺ vessels and containing mouse RBC (**Fig R2**). That way, we can confirm that those human CD31⁺ vessels were perfused with mouse RBC. Functional vessels were counted only if they contained at least 1 mouse RBC. This method was previously used by our group and others to confirm the anastomosis of human vessels with mouse vessels.⁹⁻¹¹ The authors recognize that this method is not perfect. Therefore, we also performed a perfusion study with *UEA-1* lectin and *GS-IB4* iso-lectin dyes that can specifically label the human and mouse vessels, respectively.^{12,13} Since the dyes were injected through retro-orbital veins, only the vessels that are connected to the host circulation were stained with the dyes (**Fig 6I**). We hope that using these two methods: IHC and perfusion studies, we can convincingly confirm that the implanted human ECFCs can form vessels that were connected to the mouse vessels. We have included the

detailed quantification method as part of the **Materials and Method**, as well as **Supplementary Figure 10** in the revised manuscript.

4. Figure 3 layout is not logical. Perhaps placed A-C on one row with D and E below would be more logical? Also, should the authors draw lines between paired samples in Figure 3E since the authors state that there is a connection between TAGLN expression and degree of GDM (presumed GTT)? Demonstrating pairs would also be helpful since some GDM-ECFC express similar TAGLN under both control and treatment conditions (e.g Lanes 1 and 2; Lanes 4 and 5).

We appreciate the feedback and already made changes as we understood that reflected the reviewers' comments (**Fig 3**). To demonstrate pair between TAGLN expression and degree of GDM (**Supplementary Table 1**), individual data sets in (**Fig R3E**) are color-coded (purple, yellow, blue, and red) to match the corresponding western blot bands.

5. Please use angiogenesis or vasculogenesis but not both (e.g. Figure 5)

Thank you for pointing this out. For consistency, the term **vasculogenesis** is used to define tube formation assays that resemble the early *de novo* vessel formation (e.g., when endothelial cells are evenly distributed in collagen or fibrin gels to form primitive vessel networks).¹⁴ We have corrected this terminology throughout the manuscript.

6. Line 182 reads "for at least after 6 days in culture". Please correct.

The sentence has been corrected to read "for at least 6 days in culture."

Reviewer #2:

This manuscript by Bui et al. presents the studies on the application of small molecule SB-431542 loaded liposomal nanoparticles directly conjugated to the surface of GDM exposed ECFCs to rejuvenate these circulating vascular progenitor cells. Their results supported the conclusion that controlled delivery of SB-431542 to GDM-ECFCs normalized transgelin (TAGLN) expression and improved cell migration, and eventually enhanced in vitro and in vivo vasculogenesis of GDM-ECFCs. Overall, the study presents a novel approach for improving vasculogenic potential of GDM-exposed ECFCs using TGF- β inhibitor that can potentially overcome the challenges associated with growth factor delivery or gene therapy. The study is carefully developed, and the manuscript is well-organized. However, some revisions and clarifications are required to strengthen the data and improve the quality of the manuscript, which are listed below:

We thank the reviewer for the critical reading and thoughtful critique of our manuscript. We are very excited that the reviewer found our study interesting. We have attempted to address all comments and concerns as described in detail below. We believe that the manuscript is greatly enhanced because of the constructive feedback from the reviewers.

1. The attachment of nanoparticles on the surface of ECFCs as well as cell viability was studied using nanoparticles without SB incorporation. Considering the hydrophobic nature of small molecule SB-431542, the authors are suggested to perform studies to investigate the nanoparticles conjugation and cell viability using SB incorporated nanoparticles.

The reviewer raised a very important point here. We have addressed this with the following cell viability experiment, as suggested by the reviewer. SB-431542 containing NPs were conjugated onto the surface of ECFCs with various cell to NP ratios (1:100, 1:500, 1:1000, 1:2500, and 1:5,000). After 48 hours of culture, ECFCs were evaluated using Live/Dead viability assay (i.e., Calcein AM and Ethidium homodimer-1). Representative images indicate cell viability of unconjugated ECFCs control and ECFCs with various cell to NP ratios (**Fig R4A**). Compared to unconjugated ECFCs control, we did not observe any significant differences in cell viability among groups with varying cell to NP ratios (**Fig R4B**). This result is consistent with previous studies that used SB-431542 to culture and expand vascular progenitor cells from human induced pluripotent stem cells.^{15,16} We have incorporated this additional data into **Supplementary Figure 4**.

2. The proposed drug delivery approach for rejuvenating GDM-ECFCs is based on the conjugation of the lipid particles to free thiols on the surface of GDM-ECFCs. How the conjugation of these nanoparticles to the thiol groups on the surface of other cell types including mononuclear cells or other vascular cells might adversely affect the application of SB-NPs for the intended application?

We thank the reviewer for highlighting this important point. Following conjugation of the NPs to the surface of GDM-ECFCs, the free maleimide groups on the NPs were PEGylated. Therefore, these NPs can no longer bind to the free thiols on the surface of other mononuclear or vascular cells. The intended application of this platform technology is to be used in the current clinical setting, where stem and progenitor cells are being processed *ex vivo* prior to use. We have included this discussion in the revised manuscript to clarify this interesting point raised by the reviewer.

3. In order to study the efficacy of SB-NPs *in vivo*, GDM-ECFCs were pre-conjugated with SB-NPs and encapsulated in the gels and then transplanted subcutaneously. How the authors would justify the application of their developed drug delivery system for clinical application? What is the chance of SB-NPs being up taken by circulating GDM ECFCs if the nanoparticles are administered locally or systemically?

We thank the reviewer for highlighting this important point. The authors completely understand the reviewer's concern that if the non-PEGylated NPs were administered locally or systemically *in vivo*, they will non-specifically bind to many circulating cells. However, direct injection of the NPs *in vivo* has never been the intended application of this platform technology. Instead, this NPs technology was designed specifically to be used in the current clinical setting, where stem and progenitor cells are isolated and processed *ex vivo* prior to direct transplantation or storage for future use. Since these stem and progenitor cells are already being processed *ex vivo*, it makes logical sense to add this simple and benign NPs conjugation process in order to rejuvenate them prior to transplantation. The authors envision that the rejuvenated stem cells can be used for direct transplantation or as part of engineered tissues. The study used encapsulated hydrogels to demonstrate the visibility of approach in the context of engineered constructs. We have included this discussion in the revised manuscript to clarify this interesting point raised by the reviewer.

4. Please provide the raw data and detailed statistical analysis for the figures listed below as supplemental data:

Figure 3B and 3E
Figure 4B and 4E
Figure 5C and 5F
Figure 6G and 6H

We have included the raw data and detailed statistical analysis for the above figures in the supplementary information and an attached excel file to this manuscript.

5. The authors are recommended to remove the “#” symbol used to show the statistical insignificance and only use the symbols where the comparison is statistically significant to eliminate the chance of any confusion.

We have removed the “#” symbol throughout the manuscript to eliminate any confusion.

6. What kind of CD31 mouse specific antibody was used in the study? Please specify it in the methods section.

Anti-mouse CD31 (PECAM-1) from Santa-Cruz (SC1506) with dilution factor of 1:100 was used for this study. This information together with other antibodies used for the study has been included as part of **method section** and **Supplementary Table 3**.

7. Please explain the *in vivo* perfusion procedure in the respective methods section.

The *in vivo* perfusion study was performed following an established protocol reported by our groups and others. Briefly, the perfusion study was performed on NOD/SCID mice on day 14 following the *in vivo* transplantation. Rhodamine-conjugated *UEA-1* (Vector Laboratory, dilution ratio 1:2) and FITC-conjugated *GS-IB4* solutions (Vector Laboratory, dilution ratio 1:2) were injected through retro-orbital veins. After 20 mins, intravital images were taken using a dissecting microscope (Leica, magnification 12X) and multiphoton microscope (Olympus). Confocal image stacks were acquired to create 3D rendering images, which were quantified for percent area covered by human and mouse vessels, as well as for distribution of vessel diameters using ImageJ. At the end of the experiment, the grafts were harvested, fixed with 4% paraformaldehyde, and processed for H&E staining and IHC analysis (**Supplementary Fig. 12**). Images for IHC analysis were acquired with a brightfield microscope (Revolve, Echo microscope, 10X magnification). This detailed procedure has been included as part of the **method section** and the **Supplementary Information**.

Reviewer #3:

The authors study the ability of drug-loaded nanoparticles to restore the activity of endothelial colony forming cells derived from infants born to mothers with gestational diabetes. These cells may be useful for therapeutic angiogenesis to prevent development of cardiovascular diseases later in life. The study is based on the authors' previous work demonstrating attenuation of elevated TAGLN expression can restore function of endothelial cells derived from this population. In this study, nanoparticles are bound to the cell surface to deliver a signaling molecule known to inhibit TAGLN expression. The authors demonstrate improved migration and vasculogenesis using *in vitro* and *in vivo* assays. Use of blood samples from infants born to mothers with gestational diabetes is a strength of this study due to the clinical relevance. While the data convincingly demonstrate improved functionality of the endothelial cells, their therapeutic benefit for preventing diseases that manifest over the course of decades is uncertain.

We thank the reviewer for the critical reading and thoughtful critique of our manuscript. We are very appreciative that the reviewer found our study interesting. We have attempted to address all comments and concerns as described in detail below. We believe that the manuscript is greatly enhanced because of the constructive feedback from the reviewers.

1. There are several typos that need to be corrected. Line 35 should read "restore endothelial function." Line 137 should read "without affecting." Line 327 should read "without the anticipated vascular and immune effects." Line 362 should read "the increase in both human." Line 363 should read "indicate functional chimeric vascular networks."

Thank you for bringing this to our attention and we apologize for the oversight. We have corrected the typos.

2. The authors should provide statistics to demonstrate the significance of the clinical problem. What is the incidence of gestational diabetes? What is the incidence of cardiovascular disease for individuals exposed to gestational diabetes versus the general population?

We thank the reviewer for highlighting this important point. As part of the IU Precision Health Initiative, our clinical collaborators at Indiana University School of Medicine (IUSM) are performing “The Hoosier Mom Study” to discover the causes of GDM and stop its transition into T2DM. In general, GDM affects 6% of pregnant women, but this percentage can be up to 15% in certain racial groups (i.e, Asians and African Americans).^{17,18} Intrauterine exposure to diabetes also increases the risk of cardiovascular disease (CVD) in adolescence and early adulthood. A recent study indicates that the hazard of CVD end points was elevated in offspring exposed to GDM (adjusted HR 1.42, 95% CI 1.12-1.79).¹⁹ A similar association was also observed for CVD risk factors.²⁰ We have included this additional clinical insight into the revised manuscript.

3. How do the authors envision their nanoparticle-conjugated cell therapeutic will be used clinically? Would this be a one-time treatment or a recurring treatment? Would the cells be injected intravenously? When and how often? Is improved angiogenesis at 2 weeks post treatment expected to confer a lifetime of protection against elevated risk of cardiovascular disease?

The reviewer raised a very interesting point here. This NPs technology was designed specifically to be used in the setting of current clinical practice. For stem cell therapy, currently stem and progenitor cells are isolated and processed *ex vivo* prior to direct transplantation or storage for future use. Since these stem and progenitor cells are already being processed *ex vivo*, it makes logical sense to add this simple and benign NPs conjugation process in order to rejuvenate them prior to transplantation. The authors envision that the rejuvenated stem cells can be used for direct transplantation and as part of engineered tissues.

Direct transplantation / injection can be used for repairing damaged vasculatures (revascularization), for instance in the case of peripheral artery disease / critical limb ischemia (PAD/CLI). In this case, current clinical trial requires injection of ECFCs every 3-6 months. ECFCs can also be used as part of engineered tissue to regenerate the vasculature. This can be part of tissue engineered constructs. The study used encapsulated hydrogels to demonstrate the visibility of approach in the context of engineered constructs. In this context, 2 weeks are a typical time point used to analyze the vascularization of a tissue-engineered constructs.

The authors do not intend to claim that this treatment can be used to confer a lifetime of protection against elevated risk of cardiovascular risk. Rather, the authors envision to use this NPs technology to improve stem cell function in the event where stem cell transplantation or engineered tissues can be useful to address cardiovascular complications. We have included this discussion in the revised manuscript to clarify this interesting point raised by the reviewer.

4. Would all babies born to mothers with gestational diabetes be candidates to receive and potentially benefit from this treatment? Or would there be a screening and selection process to identify suitable candidates? If so, based on what criteria?

We thank the reviewer for highlighting this important point. The authors envision that there will be selection criteria for candidates to receive and potentially benefit from this treatment. Our results demonstrate that this therapeutic strategy can normalize TAGLN expression and improve vasculogenic potential, especially for GDM-ECFCs with relatively high TAGLN expression (**Figure 3D-E**). This suggests that patients with high expression of TAGLN would benefit the most from this therapeutic strategy. The authors envision that we can use TAGLN as biomarker to define the threshold point where the clinical benefits of this therapeutic strategy could potentially outweigh the cost and risk. We have incorporated this discussion into the revised manuscript.

5. The authors state 5,000 particles per cell is the maximum that could be conjugated without affecting proliferation and viability. However, it does not appear that higher doses were studied. How can the authors conclude this is the maximum?

We thank the reviewer for the critical reading of our data. Based on previous studies, we estimate that 5% is the maximum surface coverage that we can achieve without altering cell viability and key ECFC phenotypes. Given that an average diameter of ECFCs is 15-20 microns and the size of the nanoparticles is 0.15 microns in diameter. We estimate that the nanoparticles will be sterically hindered at ratios beyond 5,000 NPs per cell. Our preliminary experiments support this calculation. The fluorescence signal from Cells: NPs at the ratio of 1:2500 and 1:5000 is when the signal starts to deviate from linearity (**Fig 1F**). Considering the NPs are ~0.15 microns in diameter, there is a possibility that at cell-NPs ratios of 1:2500 and 1:5000 there might be the fusion of NPs on the cell surface, after conjugation to the cell membrane, due to diffusion and increased steric packing of the NPs on the cell membrane. For this reason, we decided not to investigate cell-NPs ratios beyond 1:5000. We have incorporated this discussion in the revised manuscript.

6. Lines 183-184 state protein expression correlates well with gene expression. This statement is too strong as protein expression did not decrease to nearly the same extent as gene expression (Fig. 3C vs. 3E).

We thank the reviewer for critical reading of our data. We have toned down the data interpretation. The reviewer correctly pointed out that not all of the GDM-ECFC samples responded the treatment with the same rate. This is partly due to biological variability between samples and the severity of the GDM exposure. We have modified Fig 3D-E with color coding to demonstrate the different biological samples. Although PCR seems to be more sensitive to detect the differences than Western Blot, it is important to note that the functional assays (i.e., cell migration, vasculogenic assays) seem to indicate functional improvement in GDM-ECFCs following conjugation with NPs. We have included this discussion in the revised manuscript to clarify this interesting point raised by the reviewer.

7. How were mouse erythrocytes identified in chimeric vessels? Based on morphology?

Yes, mouse erythrocytes (RBC) were identified in chimeric vessels based on their unique morphology. We demonstrated some representative images of vessels with various shapes and sizes that were analyzed for human CD31⁺ vessels and containing mouse RBC (**Fig R2**). That way, we can confirm that those human CD31⁺ vessels were perfused with mouse RBC. Functional vessels were counted only if they contained at least 1 mouse RBC. This method was previously used by our group and others to confirm the anastomosis of human vessels with mouse vessels.⁹⁻¹¹ We have included the detailed quantification method as part of the **Materials and Method**, as well as **Supplementary Figure 10** in the revised manuscript.

8. In Figure 1H, why does higher drug concentration result in reduced release rate?

We thank the reviewer for highlighting this intriguing point. Non-Fickian diffusion has been observed in other drug-liposomes formulations as well. For example, in previous study vincristine sulfate was encapsulated in liposomes at a drug to lipid ratios of 1:10, 1:5, and 1:2. When release profiles of vincristine sulfate was compared for each formulation, the 1:10 drug to lipid ratio formulation had the highest release rate at 12h at 37C followed by 1:5, and 1:2 drug to lipid ratio formulations.²¹ Therefore, what we observed was not without precedent. SB 431542 is an amphiphilic molecule with polar and hydrophobic properties that are pH and solvent-dependent. The amine group on SB 431542 allows approximately 0.5 mg/mL of the molecule to be dissolved in 1:1 solution of DMSO and PBS (pH 7.2). We hypothesize that the amine group on the SB 431542 gets a partial positive charge at physiological pH. This would electrostatically interact with the phosphate groups in the lipid membrane of the liposomes. As the concentration of SB 431542 increases, it will lead to more ordered self-assembly of the drug molecule at the lipid membrane using its hydrophobic structure and amine group, which in turn possibly contributes to the increased stability of the lipid membrane.²² This in turn might contribute to the slower release of drug molecules at higher SB 431542 to lipid ratios as seen in our studies. As seen from past peer-reviewed publications this phenomenon is not uncommon for small molecule amphiphilic drugs and lipid formulations. We have included this discussion in the revised manuscript to clarify this interesting point raised by the reviewer.

9. Figure 1C and 1D, please add units to the vertical axes.

The units have been added to Figure 1C and 1D.

References

1. Méndez-Ferrer S, Scadden DT, Sánchez-Aguilera A. Bone marrow stem cells: current and emerging concepts. *Annals of the New York Academy of Sciences*. 2015;1335(1):32–44.
2. D'Souza A, Fretham C, Lee SJ, et al. Current Use of and Trends in Hematopoietic Cell Transplantation in the United States. *Biology of Blood and Marrow Transplantation*. 2020;26(8):e177–e182.
3. Stephan MT, Moon JJ, Um SH, Bershteyn A, Irvine DJ. Therapeutic cell engineering with surface-conjugated synthetic nanoparticles. *Nature Medicine*. 2010;16:1035.
4. Stephan MT, Irvine DJ. Enhancing cell therapies from the outside in: Cell surface engineering using synthetic nanomaterials. *Nano Today*. 2011;6(3):309–325.
5. Tsirikis P, Wilson K, Xiang S, et al. Immunogenicity and biodistribution of nanoparticles *in vivo*. *J. Immunol*. 2016;196(1 Supplement):75.28.
6. Reid E, Guduric-Fuchs J, O'Neill CL, et al. Preclinical Evaluation and Optimization of a Cell Therapy Using Human Cord Blood-Derived Endothelial Colony-Forming Cells for Ischemic Retinopathies. *Stem Cells Transl Med*. 2018;7(1):59–67.
7. Patschan D, Schwarze K, Tampe B, et al. Endothelial Colony Forming Cells (ECFCs) in murine AKI - implications for future cell-based therapies. *BMC Nephrol*. 2017;18(1):53–53.
8. Alphonse RS, Vadivel A, Zhong S, et al. The isolation and culture of endothelial colony-forming cells from human and rat lungs. *Nature Protocols*. 2015;10(11):1697–1708.
9. Mead LE, Prater D, Yoder MC, Ingram DA. Isolation and Characterization of Endothelial Progenitor Cells from Human Blood. *Current Protocols in Stem Cell Biology*. 2007;
10. Hanjaya-Putra D, Bose V, Shen Y-I, et al. Controlled activation of morphogenesis to generate a functional human microvasculature in a synthetic matrix. *Blood*. 2011;118(3):804 LP – 815.
11. M. M-MJ, E. DOM, Soo-Young K, et al. Engineering Robust and Functional Vascular Networks In Vivo With Human Adult and Cord Blood-Derived Progenitor Cells. *Circulation Research*. 2008;103(2):194–202.
12. Cheng G, Liao S, Kit Wong H, et al. Engineered blood vessel networks connect to host vasculature via wrapping-and-tapping anastomosis. *Blood*. 2011;118(17):4740–4749.
13. Hanjaya-Putra D, Shen Y-I, Wilson A, et al. Integration and Regression of Implanted Engineered Human Vascular Networks During Deep Wound Healing. *STEM CELLS Translational Medicine*. 2013;2(4):297–306.
14. Davis GE, Stratman AN, Sacharidou A, Koh W. Molecular basis for endothelial lumen formation and tubulogenesis during vasculogenesis and angiogenic sprouting. *Int Rev Cell Mol Biol*. 2011;288:101–165.
15. James D, Nam H, Seandel M, et al. Expansion and maintenance of human embryonic stem cell-derived endothelial cells by TGFbeta inhibition is Id1 dependent. *Nat Biotechnol*. 2010;28(2):161–166.
16. Prasain N, Lee MR, Vemula S, et al. Differentiation of human pluripotent stem cells to cells similar to cord-blood endothelial colony-forming cells. 2014;32:1151.

17. Shah NS, Wang MC, Freaney PM, et al. Trends in Gestational Diabetes at First Live Birth by Race and Ethnicity in the US, 2011-2019. *JAMA*. 2021;326(7):660–669.
18. Lawrence RL, Wall CR, Bloomfield FH. Prevalence of gestational diabetes according to commonly used data sources: an observational study. *BMC Pregnancy and Childbirth*. 2019;19(1):349.
19. Guillemette L, Wicklow B, Sellers EAC, et al. Intrauterine exposure to diabetes and risk of cardiovascular disease in adolescence and early adulthood: a population-based birth cohort study. *CMAJ*. 2020;192(39):E1104.
20. Yu Y, Arah OA, Liew Z, et al. Maternal diabetes during pregnancy and early onset of cardiovascular disease in offspring: population based cohort study with 40 years of follow-up. *BMJ*. 2019;367:l6398.
21. Mao W, Wu F, Lee RJ, Lu W, Wang J. Development of a stable single-vial liposomal formulation for vincristine. *Int J Nanomedicine*. 2019;14:4461–4474.
22. Fahr A, Hoogevest P van, May S, Bergstrand N, S. Leigh ML. Transfer of lipophilic drugs between liposomal membranes and biological interfaces: Consequences for drug delivery. *European Journal of Pharmaceutical Sciences*. 2005;26(3):251–265.

REVIEWERS' COMMENTS:

Reviewer #1 (Remarks to the Author):

I appreciate the authors' considerable effort to answer the questions of all reviewers. The revised manuscript is complete and a helpful addition to the literature. I recommend acceptance.

Reviewer #2 (Remarks to the Author):

The authors sufficiently addressed the reviewer's comments.

Reviewer #3 (Remarks to the Author):

The authors have addressed the reviewer concerns to the satisfaction of this reviewer.